# Evaluation of modelling $NO_2$ concentrations driven by satellite-derived and bottom-up emission inventories using in-situ measurements over China

Fei Liu[1,2,3], Ronald J. van der A[1], Henk Eskes[1], Jieying Ding[1,4], Bas Mijling[1]

[1]Royal Netherlands Meteorological Institute (KNMI), Department of Satellite Observations, De Bilt, the Netherlands
[2]Universities Space Research Association (USRA), GESTAR, Columbia, MD, USA
[3]NASA Goddard Space Flight Center, Greenbelt, MD, USA
[4]Department of Geoscience and Remote Sensing (GRS), Delft University of Technology, Delft, the Netherlands

*Correspondence to*: Fei Liu (fei.liu@nasa.gov; liuf1010@gmail.com)

**Abstract.** Chemical transport models together with emission inventories are widely used to simulate $NO_2$ concentrations over China, but validation of the simulations with in situ measurements has been extremely limited. Here we use ground measurements obtained from the air quality monitoring network recently developed by the Ministry of Environmental Protection of China to validate modelling surface $NO_2$ concentrations from the CHIMERE regional chemical-transport model driven by the satellite-derived DECSO and the bottom-up MIX emission inventories. We applied a correction factor to the observations to account for the interferences of other oxidized nitrogen compounds ($NO_z$), based on the modelled ratio of $NO_2$ to $NO_z$. The model accurately reproduces the spatial variability of $NO_2$ from in-situ measurements, with a spatial correlation coefficient of over 0.7 for simulations based on both inventories. A negative and positive bias is found for the simulation with the DECSO (slope = 0.74/0.64 for the daily-mean/daytime only) and the MIX (slope = 1.3/1.1) inventory respectively, suggesting an underestimation and overestimation of $NO_x$ emissions from corresponding inventories. The bias between observed and modelled concentrations is reduced with the slope dropping from 1.3 to 1.0 when the spatial distribution of $NO_x$ emissions in the DECSO inventory is applied as the spatial proxy for the MIX inventory, which suggests an improvement of the distribution of emissions between urban and suburban/rural areas in the DECSO inventory compared to that used in the bottom-up inventory. A rough estimate indicates that the observed concentrations, from sites predominantly placed in the populated urban areas, may be 10–40% higher than the corresponding model grid-cell mean. This reduces the estimate of the negative bias of the DECSO based simulation to the range of -30% to 0% on average, and establishes more firmly that the MIX inventory is biased high over major cities. The performance of the model is comparable over seasons, with a slightly worse spatial correlation in summer, due to the difficulties in resolving the more active $NO_x$ photochemistry and larger concentration gradients in summer by the model. In addition, the model well captures the daytime diurnal cycle, but shows more significant disagreement between simulations and measurements during night time, which likely produces a positive model bias of about 15% in the daily mean concentrations. This is most likely related to the uncertainty in vertical mixing in the model at night.

## 1 Introduction

Nitrogen dioxide (NO$_2$) is an important trace gas in the troposphere. It actively participates in the formation of tropospheric ozone and secondary aerosols (Seinfeld and Pandis, 2006), which influences human health and impacts climate significantly. Emissions of NO$_2$ together with nitric oxide (NO) that is rapidly converted to NO$_2$ in the troposphere during daytime are closely related to anthropogenic activities, in particular fossil fuel consumption, which has increased global NO$_x$ (NO +NO$_2$) emissions by a factor of 3–6 since preindustrial times (Prather et al., 2001). China is one of the largest contributors to NO$_x$ emissions over the world, contributing to 18% of global NO$_x$ emissions based on the estimate of EDGAR v4.2 (European Commission (EC): Joint Research Centre (JRC)/Netherlands Environmental Assessment Agency (PBL), 2011), as a consequence of the large energy consumption driven by the rapidly growing economy. A good understanding of NO$_2$ levels as well as temporal and spatial variations is urgent to help solve the serious environmental problems, particularly poor air quality, caused by emissions.

Chemical transport models (CTMs) have been widely used to provide predictions of gas phase pollutants including NO$_2$ and particulate matter concentrations, which are powerful tools for understanding regional air pollution issues, assessing emission control scenarios (Kiesewetter et al., 2014) and analysing trans-boundary transport (Streets et al, 2007). The modelled NO$_2$ concentrations have received extensive evaluation by comparing with ground-based measurements (Pay et al., 2012), satellite observations (Huijnen et al., 2010) and airborne observations (Carmichael et al., 2003) for regional (Stern et al., 2008) and urban-scale (Terrenoire et al., 2015) air quality simulations. The results of these intercomparisons show quite good performance of the models, but still suggest uncertainties in the estimation of the meteorological input data (Bessagnet et al., 2016), the modelling of NO$_x$ chemistry (Valin et al., 2011) and particularly emission inventories (A. Mues et al., 2014). Emission inventories are necessary input to CTMs and recognized as one of the most important sources of uncertainties. Traditional bottom-up emissions are calculated by aggregating information from diverse sources of information such as fuel statistics and measurements of emission factors. The large uncertainties in energy statistics (Guan et al., 2012) and applications of non-Chinese emission factors (Streets et al., 2003) have been propagated into uncertainties in bottom-up inventories for China (Zhao et al., 2011). The lack of bottom-up inventories for most recent years introduces additional biases for model simulations, because inventories could quickly become outdated due to the rapidly changing emissions (Zhang et al., 2007; Liu et al., 2016a). NO$_2$ columns detected from space provide additional constraints to yield a satellite-derived NO$_x$ emission inventory. Initially, NO$_x$ emissions have been estimated from satellite observations together with CTMs at coarse resolution, based on the assumption of a linear relationship between NO$_2$ columns and NO$_x$ emissions ignoring pollution transport (Martin et al., 2003). More complicated techniques like the Kalman filter (Napelenok et al., 2008) and four-dimensional variational data assimilation (4DVAR) (Kurokawa et al., 2009) have been introduced to take pollution transport into account. In addition, CTM-independent methods have been developed for point sources (Beirle et al., 2011;

Liu et al., 2016b). The uncertainties in $NO_2$ columns retrievals (Dirksen et al., 2011), in particular for China with high loadings of aerosols (Ma et al., 2013), together with estimation method uncertainties, result in errors in satellite-derived inventories.

The modelling of $NO_2$ concentrations over China has been evaluated with space- and ground-based observations. Reported validation studies have focused on evaluating tropospheric $NO_2$ column densities simulated by CTMs driven by bottom-up emission inventories using satellite measurements. Differences between the simulated $NO_2$ column densities and observations of the Global Ozone Monitoring Experiment (GOME) (Ma et al., 2006; Uno et al., 2007), the Scanning Imaging Absorption Spectrometer for Atmospheric CHartographY (SCIAMACHY) (Shi et al., 2008) and the Ozone Monitoring Instrument (OMI) (Wang et al., 2011) have been attributed to uncertainties in the magnitude and spatial distribution of bottom-up emissions. The validation of surface $NO_2$ concentrations was generally performed for limited time periods using a limited set of measurement stations (e.g., three large cities in Wang et al. (2011), one or two sites in Wang et al. (2007) and the City of Nanjing in Ding et al. (2015)), due to the absence of routine monitoring data. Alternatively, the satellite-derived inventories were compared to bottom-up inventories directly, which shows considerable disparity (Lin et al., 2012; Ding et al., 2017a).

Measurements obtained from the recently developed air quality monitoring network in China (Zhang and Cao, 2015) provide the means to evaluate the quality of $NO_2$ modelling. We evaluate the surface $NO_2$ concentrations simulated by a CTM driven by both satellite-derived and bottom-up inventories with this newly-established dataset. To our knowledge, this is the first time that modelled $NO_2$ concentrations over China have been evaluated with in situ measurements throughout the country, while an intercomparison for simulations with satellite-derived and bottom-up inventories is performed simultaneously. We structure the paper as follows. In section 2.1 and 2.2 the CTM and emission inventories adopted in this study are described respectively. The introduction of the in situ measurements from the air quality monitoring network in China and the correction for interference of in situ $NO_2$ data are given in section 2.3 and 2.4 respectively. Annual mean simulated surface $NO_2$ are compared with the corrected in situ measurements in Section 3.1. Further analysis focusing on seasonality and diurnal cycle are provided in Section 3.2 and 3.3 respectively. Section 4 presents a summary of the major findings in this paper.

## 2 Methodology

### 2.1 CHIMERE model

We used the CHIMERE regional chemical-transport model in this study, which is designed to produce daily forecasts of tropospheric trace gas and aerosol pollutants and make long-term simulations at a range of spatial scales (Menut et al., 2013). We use the CHIMERE model v2013b over East Asia (18°N to 50°N and 102°E to 132°E) with a resolution of 0.25° following the configuration in Ding et al. (2015). The CHIMERE simulation was driven by operational meteorological data from the European Centre for Medium-Range Weather Forecasts (ECMWF) with a horizontal resolution of 0.25°.

Atmospheric variables were simulated in 8 layers from the surface to 500 hPa. Tropospheric photochemistry is represented using the reduced MELCHIOR chemical mechanism (Derognat et al., 2003), including about 120 reactions and 44 gaseous species. Aerosol module accounting both for inorganic and organic species of primary or secondary origin is included according to Bessagnet et al. (2004). Boundary conditions for the model domain were derived from monthly mean

climatology based on Model for OZone And Related chemical Tracers (MOZART) second-generation (Horowitz et al., 2003) for gases, the Laboratoire de Météorologie Dynamique Zoom – Interaction avec la Chimie et les Aérosols (LMDz-INCA; Folberth et al., 2006) for nitrate and ammonium, and the Georgia Tech/Goddard Global Ozone Chemistry Aerosol Radiation and Transport (GOCART, Ginoux et al., 2001) for other aerosols. At default, $NO_x$ emissions are speciated as 9.2% of $NO_2$, 0.8% of HONO and 90% of NO in the CHIMERE model (Menut et al., 2013), following the Generation of European

Emission Data for Episodes (GENEMIS) recommendations (Friedrich, 2000; Kurtenbach et al., 2001; Aumont et al., 2003). Open access satellite-derived and bottom-up inventories that provide up-to-date emissions over East Asia were selected to drive the model in this study, which will be detailed in Sect. 2.2.

## 2.2 Emission inventory

The satellite-derived $NO_x$ emissions were estimated by the algorithm DECSO (Daily Emission estimates Constrained by

Satellite Observations) v5 using an extended Kalman filter (Mijling and van der A, 2012; Ding et al., 2015; Ding et al., 2017b). DECSO uses one forward model run of a CTM to calculate the response of $NO_2$ concentrations to both local and non-local $NO_x$ emissions. Daily OMI $NO_2$ observations retrieved with the DOMINO version 2 algorithm (Boersma et al., 2011) are used as a constraint to update emissions. The DECSO emission data are available at www.globemission.eu.

The bottom-up anthropogenic $NO_x$ emissions were taken from the MIX inventory (Li et al., 2017a), a mosaic Asian

anthropogenic emission inventory under the international collaboration framework of the Model Inter-Comparison Study for Asia (MICS-Asia) and the Task Force on Hemispheric Transport of Air Pollution (TF HTAP). The MIX inventory is developed for the years 2008 and 2010 by an integration of state-of-the-art regional emission inventories for all major anthropogenic sources in 29 countries and regions over Asia. The emissions of China integrated in the MIX inventory are derived from the Multi-resolution Emission Inventory for China (MEIC: http://www.meicmodel.org) compiled by Tsinghua

University. The anthropogenic emissions together with the biogenic emissions, which were computed automatically in the CHIMERE model using the global MEGAN (Model of Emissions of Gases and Aerosols from Nature) model (Guenther et al., 2006), were adopted as bottom-up inventory. We refer to this combination as the MIX inventory for brevity hereinafter. Note that monthly emissions for all above inventories were provided at the spatial resolution of $0.25° \times 0.25°$.

Both inventories show comparable spatial distributions at a national and regional scale, but distinctions between urban and

rural areas (see Sect.3.1). The strength of the MIX inventory is that it includes detailed source-category information (e.g., power plant and transportation sector) for emissions, which is useful for driving atmospheric models and designing emission mitigation policies, but not included in DECSO. The advantage of the DECSO inventory is that emissions are timely updated (as soon as the satellite observations are available); while bottom-up inventories usually lag behind a few years and are

outdated by the time they become available. In addition, the spatial information in DECSO is based on OMI $NO_2$ observations, while MIX relies on spatial proxies like GDP to allocate emissions due to the lack of data. An in-depth comparison between inventories has been described by Ding et al. (2017).

We focused on 2015 as the most recent year with available DECSO emission estimates and in situ measurements, but we used the MIX inventory for 2010, because the year 2015 is not available yet. However, the use of the 2010 MIX inventory without scaling is not expected to bring significant bias, as the similarity of $NO_x$ emissions for 2010 and 2015 has been reported by both the bottom-up inventory MEIC (Liu et al, 2016a) and the satellite-derived inventory DECSO. For the period of 2010–2012, the $NO_x$ emissions of China experienced a rapid growth. A sharp decline of $NO_x$ emissions was observed in the years of 2013–2015, with a peak around 2012 (Liu et al., 2016a). As a result, the inventory for 2010 is comparable to that for 2015, even though there is a five-year lag. Figure 1 compares DECSO $NO_x$ emissions for 2015 (left) and 2010 (middle), which are consistent in both total amount (21.5 vs 21.6 Tg) and spatial distribution (r= 0.83). Figure 1 further displays the spatial distributions of the MIX $NO_x$ emissions for 2010 (right). These emissions are significantly higher (39%) than the DECSO inventory when averaged over the model domain.

An air quality simulation using the CHIMERE model was conducted for the full year 2015. Pollutant concentrations including $NO_2$ were simulated based on the 2015 DECSO and the 2010 MIX $NO_x$ inventories respectively. Note that 2010 MIX inventory for other species was used together with both $NO_x$ inventories. Because of the inconsistency between the emission sectors used in the DECSO and the MIX inventories and that in SNAP (Selected Nomenclature for Air Pollution) 97, which are internally used in the CHIMERE model, we adopted the sector mapping table as discussed in Ding et al. (2015). The concentration in the lowest model layer (from the ground up to 20 m) was used for validation against surface $NO_2$ observations in this study. Figure 2 illustrates the annual mean surface $NO_2$ simulation using both inventories. Large enhancements are found over industrial regions, in particular northern China, North China Plain and Yangtze River Delta. The model run based on the MIX inventory (Fig. 2, middle) shows overall larger concentrations than that based on the DECSO inventory (Fig. 2, left).

## 2.3 Ground-Level in situ measurements

The real-time hourly $NO_2$ concentrations as well as other major air pollutants are continuously recorded by the Ministry of Environmental Protection (MEP) in China and are publicly accessible from the year 2013 onwards (Zhang and Cao, 2015). We obtained the hourly in situ measurements from a total of 1413 air quality monitoring sites of the MEP network for 323 major cities over the model domain. A majority of those monitoring sites has been placed in the city center, and are named urban assessing stations in the official document (MEP, 2013). These are meant to evaluate the overall level and trend of air quality for areas with the highest concentrations and highest population exposure. The placement criteria of urban assessing stations laid down in the legislation (MEP, 2013) ensure that the measurements are representative for urban areas. Stations are required to be well-distributed within the developed area of the city and not too close to stationary emission sources (50 m) or roads (10–100 m depending on the traffic flow). The minimum number of monitoring sites required per city depends

on both the urban population and city size, i.e., at least one station for an area of ~50 km$^2$ (Table 1). In addition, for areas with the concentration exceeding the grade II of national ambient air quality standard (i.e., the annual mean NO$_2$ concentration of 40 μg/m$^3$; MEP, 2012), the minimum required number of monitoring sites is increased by 50%. MEP also operates other types of measurement sites, including regional and background stations to assess the background air pollution

levels and pollutant transport, and source impact and traffic stations close to emission sources. However, only mega cities like Beijing and Guangzhou operate such non-urban stations. The fact that urban observations are dominating should be kept in mind when comparing the observations with the model results. The horizontal resolution of the model is limited to 0.25° which will cause representativeness errors (biases) when comparing the measurements from city stations with the mean of a grid box of the simulations, which can also include rural areas. Note that only the measurements for the dates with 24-hour

valid measurements (larger than 0) are used for the following analysis in this study.

Figure 3a displays the heterogeneous spatial distribution of monitoring sites at the scale of the model grid cell. The over 1000 monitoring sites are allocated to a total of 594 grid cells based on their geolocations. The sites belonging to the grid cells with one, two and three sites account for 17%, 21% and 22% of the total respectively (Fig. 3b). We calculated the averaged distance between monitoring sites by averaging individual pairwise distances for every two stations in the same

grid cell. Because most monitoring sites are urban stations and are clustered in the city areas, which are often much smaller than the area of a grid cell (~600 km$^2$), the averaged distance is rather small with an average of 3.6 km for all grid cells as shown in Fig 3c. For mega cities with significantly larger built-up areas and thus more monitoring sites, the distribution of sites is more homogeneous over the grid cell and results in a lager distance between stations. The average distance increases from 5 km for grid cells with only one pair of stations to 11 km for those with over eight stations.

In our analysis, we excluded in situ measurements from cities with unexpected discrepancies between urban and suburban stations. Because only large cities potentially place the monitoring sites outside urban areas related to the rapid expansion of built-up areas, we classified stations as urban and suburban by visually inspecting satellite imagery from Google Earth for large cities with over four stations. We calculated the annual mean NO$_2$ of each station. When the NO$_2$ concentration of urban stations is less than that of suburban stations, the measurements are behaving differently from expectation. The cities

(4 in total) detected to have unexpected measurements are labelled as "unselected" and discarded from the validation dataset. Note that suburban stations presenting higher NO$_2$ levels than urban stations but close to large emission sources, e.g., industrial park and airport are understandable, and thus are not excluded from the database. Figure 4 presents the daily-average surface NO$_2$ abundance for the city of Xi'an. Only the dates with 24-hour valid measurements (lager than 0) are used for the time series illustration here. The expected enhancement in winter highlighted by both urban (red line) and suburban

(blue line) stations has not been detected for the urban station with lower annual mean NO$_2$ abundance than suburban stations (black line), which provides further support for excluding the Xi'an measurements from the model evaluation.

## 2.4 Correction factor

$NO_2$ concentrations are measured by commercial chemiluminescence analyzers (Zhang and Cao, 2015), which are subject to a systematic overestimation of ambient $NO_2$ concentrations (Steinbacher et al., 2007). $NO_2$ is catalytically transformed into NO by a molybdenum converter and subsequently measured by chemiluminescence. However, other reactive oxidized

nitrogen compounds ($NO_z$) such as peroxyacetyl nitrate (PAN) and nitric acid are also partly converted to NO, resulting in an overestimation of the measured $NO_2$.

We applied a correction factor proposed by Lamsal et al. (2008) to account for the interferences of other oxidized nitrogen compounds, based on the modelled ratio of $NO_2$ to $NO_z$. The correction factor CF was calculated from the local chemical concentrations as follows:

$$CF = \frac{NO_2}{NO_2 + \sum AN + 0.95 \times PAN + 0.35 \times HNO_3} \qquad (1)$$

where $\sum AN$ is the sum of all alkyl nitrates concentrations.

Figure 5 shows the seasonal means of the correction factors determined with concentrations of the interfering species predicted by the CHIMERE model driven by the DECSO inventory. Consistent with the findings in Europe (Huijnen et al., 2010) and the US (Lamsal et al., 2008), the correction factor (difference with the ideal value of 1.0) is largest over urban

polluted regions, where $NO_x$ is a larger fraction of total oxidized nitrogen compounds. The correction factor tends to be closer to unity in winter, when the $NO_x$ photochemistry is slower and thus $NO_x$ has a larger relative contribution to total oxidized nitrogen compounds. The correction factor derived from simulations with the MIX inventory (not shown) shows a similar pattern with Fig.5, but with a larger number of values close to 1 related to the larger emissions. Hourly correction factors for individual hour of each day during the year for all individual stations have been applied to the in situ

measurements. It is difficult to quantify the accuracy of the correction factors and errors, as the collocated measurements of other oxidized nitrogen compounds are not public available. We used the standard deviation of the daily means of correction factors within a season as a measure of its uncertainty. The average standard deviations for all sites are 10%, which are comparable to the uncertainty level pointed out by the study of McLinden et al. (2014).

## 3 Results and discussion

### 3.1 Annual intercomparison

We compare the modelled surface $NO_2$ with the corrected in situ measurements throughout China. In general, the spatial distribution of annual mean $NO_2$ concentrations from the CHIMERE model simulations is well in line with that from in situ measurements with a correlation coefficient of over 0.7. However, the modelled $NO_2$ is biased compared to ground measurements. The differences of annual mean $NO_2$ concentrations between simulations and measurements are given in Fig.

6. The CHIMERE simulations with the DECSO inventory show considerably lower $NO_2$ concentrations than the in situ measurements, with a negative difference for nearly 90% of all grid cells. On the contrary, the CHIMERE simulations with

the MIX inventory are generally higher than the in-situ measurements for grid cells corresponding to large cities: A positive bias is found for 70% of the grid cells with over four monitoring sites.

Grid cells are classified into five categories, i.e., mountainous, northern, < 4 stations, densely located and main sample, and the corresponding scatter plots of corrected measurements against simulations are shown in Figure 7. We define a grid cell as "mountainous" where the average elevation is higher than 1000 m and the standard deviation of elevations is over 15% of the mean, based on the topographic data from the 30-arc-sec global land topography "GTOPO30" archived by the U.S. Geological Survey (available at https://lta.cr.usgs.gov/GTOPO30, rescaled to 0.05°). The grid cells higher than 45°N are classified as "northern". The grid cells with less than 4 measurement stations are classified as "< 4 Stations". The grid cells with only densely located stations (see definition later in this section) are classified as "densely located". Note that the priority of the category of mountainous, northern, < 4 stations and densely located is from high to low when we do the classification in this study. For instance, for grid cells meet the criteria of both mountainous and northern, we classify them as mountainous. The remaining grid cells are classified as "main sample". The results for the daytime period (8:00–19:00) are displayed separately. The correlation coefficient, regression slope and root mean square error for the individual categories compared to measurements are given in Table 2.

Significant regional differences are found. The small slope over mountainous regions could be related to model limitations to resolve cities in the valleys. Furthermore, we may expect difficulties for the model in describing $NO_2$ concentrations over complex terrain. For mountainous regions, the lower slope may also be related to the large uncertainties in the meteorological parameters associated with the difficulties in resolving the characterization of small-scale orography in the ECMWF model (Beljaars et al., 2004). The errors on meteorological parameters, such as mixing height and temperature (Hongisto, 2005) and wind fields (Minguzzi et al., 2005), can introduce biases for air quality simulations. In addition, the accuracy of the DECSO algorithm highly relies on the appropriate wind fields, because DECSO performs trajectory analysis to account for $NO_x$ transport away from the source when calculating the sensitivity of concentrations to emissions (Mijling and van der A, 2012). In this way, uncertainties in meteorological parameters are amplified in the DECSO inventory, resulting in a worse agreement for mountainous areas (r = 0.51) compared to the MIX inventory (r = 0.77).

The CHIMERE model accurately reproduces the spatial variability of $NO_2$ for northern grid cells with a high correlation coefficient of 0.92 and 0.81 for the simulations with the DECSO and the MIX inventory respectively, but with a large negative bias. The bias could be related to model uncertainties in $NO_x$ sinks for high latitudes (Ding et al., 2017b), indicated by the sensitivity studies of modelled $NO_2$ columns to errors in chemical parameters associated with $NO_x$ sinks (Lin et al., 2012; Stavrakou et al., 2013). Additionally, the bias is particularly significant for the simulations with the DECSO inventory, showing a slope of merely 0.20. This could be further explained by the general underestimation of $NO_x$ emissions caused by the bias in $NO_2$ tropospheric columns of DOMINO v2 for this area (Ding et al., 2017b), partly due to a bias in the calculation of air mass factor for retrievals at large solar zenith angles by the radiance transfer model (Lorente et al., 2017) and possibly biases in the estimated stratospheric background.

Figure 8 depicts the ratio of the simulated annual mean surface $NO_2$ concentrations to the corrected in situ measurements sorted by the number of stations located in the same grid cell from small to large. It is interesting to note that the ratio is small for grid cells with less than 4 stations, but increases along with the increase of the number of stations from 4 to 9, ranging from 0.6 to 1.0 and 0.9 to 1.8 for the simulations with the DECSO and the MIX inventory respectively. The trend in the ratio suggests that the representativeness of in situ measurements for the average $NO_2$ levels of a grid cell improves with increasing numbers of stations or city size. For grid cells with less than four stations, the model with its limited spatial resolution cannot be expected to accurately resolve the spatial gradient of pollutants towards the city centre in relatively smaller urban areas. Similarly, the in situ measurements for stations located close together are expected to be less representative of the grid-cell mean compared to the homogeneously distributed stations. This is in agreement with the tendency that grid cells with lower average measurement station distances (< 10 km) tend to show lower ratios (< 0.5) in Fig. 8, in particular for grid cells with larger number of stations. We select the grid cells with over four stations, but lower average distances than the 10% percentiles in Fig. 3c, and name them as "densely located". We analyse the performance of the model in those grid cells with only densely clustered stations (labelled with letters "L" in Fig. 6). Not surprisingly, the simulations for those grid cells show larger discrepancies compared to the measurements, and also the correlation coefficient of simulations with the DECSO inventory drops down to a rather low value of 0.55 (Table 2).

We exclude grid cells in the special categories discussed above (i.e., mountainous, northern, < 4 stations and densely located stations) to draw conclusions on the ability of the model to reproduce the measurements. Statistical values (correlation, slope, root mean squared error) and least squares regression lines for the remaining grid cells ("main sample") are given along with the plots in Figure 7. A slope of 0.74 and 1.3 is found for the simulation with the DECSO and the MIX inventory respectively. As mentioned before, the majority of stations are located in urban, populated and polluted areas and the model resolution of 0.25° will not be enough to represent the existing $NO_2$ gradients, thus we may expect a negative representativity offset in the modelled surface concentration, even for the "main sample" obtained after data screening (Irie et al., 2012; Lin et al., 2014). We select grid cells with over 4 stations, which potentially place one or two stations in background areas, to give a rough estimate of the offset. The background stations are defined as stations located far away from urban areas on the basis of a visual inspection of satellite imagery from Google Earth. Not surprisingly, the measurement from the background station which is expected to better represent the grid-cell mean is smaller than the average value of measurements from all stations located in the same grid cell. The ratios of annual mean measurements from the background stations to the mean of corresponding measurements from all stations range from 0.64 to 0.86, with an average of 0.74. That is, the average negative representativity offset may reach 25%. The ratio is closer to 1 in winter (0.83 on average) due to the reduced spatial gradients in $NO_2$ caused by longer $NO_x$ lifetime, which will be discussed in detail in Sect. 3.2. Thus, the slopes of 0.74 and 1.3 for DECSO and MIX actually indicate a slightly negative and more significantly positive bias respectively.

A positive bias may indicate an overestimation of $NO_x$ emissions, or errors in the spatial downscaling of the bottom-up emission totals, although biases from the description of the chemistry, transport and removal processes in the model cannot

be ruled out. The overestimation of the MIX results over large cities, are consistent with previous findings that regional inventories like MIX have large positive biases in urban areas (Zheng et al., 2017). The reason for the positive biases will be discussed in detail later in this section. Uncertainties in the DECSO results may be attributed to biases in the OMI tropospheric $NO_2$ column densities, or representation errors introduced by the projection of the CTM onto the measured $NO_2$

satellite footprint (Ding et al., 2017b). OMI $NO_2$ observations have been reported to be systematically smaller than those from ground-based measurements (e.g., MAX-DOAS) over polluted regions (Shaiganfar et al., 2011; Ma et al., 2013; Ialongo et al., 2016), due to their different spatial representativeness (Irie et al., 2012; Lin et al., 2014) and uncertainties in $NO_2$ vertical column retrieval including the shielding effect of aerosols (Shaiganfar et al., 2011) and the varying observation geometry (Vasilkov et al., 2017). In addition, the fact that the adopted model resolution is not sufficient to accurately model

nonlinear effects in the $NO_2$ loss rate may contribute to the negative bias (Valin et al., 2011).

Regional bottom-up inventories tend to have large positive biases in urban areas. Those inventories usually downscale local emissions from regional totals (provincial totals are used in the MEIC/MIX inventory for China) and distribute them to grid cells using spatial proxies (e.g., population density and GDP). However, the spatial proxies may not match the locations of the individual emitting sources, especially for industrial plants located far away from urban centres that tend to have a larger

population density and GDP (Liu et al., 2017). Such a decoupling will result in an overestimation of emissions over urban areas, which has been proven by the comparison of proxy-based regional inventory with high-resolution urban inventories developed from the extensive use of information of individual emitting sources (Zheng et al., 2017).

In order to better compare the spatial distributions of the two inventories and identify the sensitivity of model performance on spatial distributions of emissions, we further evaluate the impact of the spatial distribution of emissions on simulating

$NO_2$ by applying the same spatial proxy for $NO_x$ emissions in both inventories. We scale the total amount of emissions of the 2015 DECSO inventory over the domain adopted in this study to that of the 2010 MIX inventory, but kept the DECSO spatial distribution (hereinafter as the corrected DECSO inventory, see Fig. 2). We then compare the modelled $NO_2$ using the corrected DECSO inventory with in situ measurements in Fig. 6c. It is interesting to see that many high values in the MIX simulation are not reproduced by the simulation with the corrected DECSO inventory. We further assess the simulation

results with the corrected DECSO inventory in Fig. 7c. The simulation with the MIX inventory tends to cluster the pollutants more over urban areas than that with the corrected DECSO inventory, indicating that the modelled $NO_2$ is sensitive to the spatial distribution of emissions. The large bias in the MIX inventory is reduced significantly with a slope decreasing from 1.3 to 1.0, which suggests an improvement of the distribution of emissions between urban and suburban/rural areas.

**Note that due to the lack of the 2015 inventory, the use of the 2010 MIX emissions for other species including $SO_2$, CO**
**and NMVOC in both the MIX and the DECSO simulations may introduce uncertainties to the simulating $NO_2$. The anthropogenic emissions of $SO_2$, CO and NMVOC for China have been reported to decrease by 2%, 5% and increase by 21% from 2010 to 2015, respectively (Li et al., 2017b). In gas-phase chemistry, the principal sink of $NO_x$ is oxidation to $HNO_3$. The influence of the growth in NMVOC on the oxidizing power of the atmosphere is partially compensated by the reduction in CO, as CO and hydrocarbons play similar roles in depleting oxidants following the**
**$HO_x$-$NO_x$-CO-hydrocarbon chemical mechanisms (Jacob, 2000). Additionally, $SO_2$ contributes to influence $NO_2$**

concentrations by forming aerosols, concentrations of which have impact on photolysis rates and thus photochemical reaction rates associated with $NO_x$ (Mailler et al., 2017).  However, the emission changes are rather small compared to the uncertainties in bottom-up estimates, which are even smaller than the discrepancies between estimates from different bottom-up inventories. Thus we believe the uncertainties arising from the use of 2010 inventory is not significant. A sensitive analysis will be further expected to quantify the influence of emissions of other species on simulated $NO_2$.3.2 Seasonality

Figure 9 compares the monthly mean $NO_2$ concentrations simulated by the CHIMERE model using the two inventories with the in situ measurements. The spatial correlation between the modelled $NO_2$ concentrations and the in-situ measurements shows a week dependence on season, which is slightly worse in summer (July). The correlation coefficients range from 0.64 (July) to 0.73 (January) and from 0.80 (July) to 0.83 (January), for the simulations with the DECSO and the MIX inventory, respectively. A possible explanation for the somewhat higher correlation in January is the smaller model error in winter than in summer, as indicated by previous findings in both China (Lin et al., 2012) and Europe (Huijnen et al., 2010). This may be related to the difficulties in resolving the more active $NO_x$ photochemistry in summer by the model. For instance, the model with a horizontal resolution of 0.25° is not able to fully resolve the spatial gradients of $NO_2$ close to strong emission sources, but such impact is smaller in winter than in other seasons, as the $NO_2$ gradients in a grid cell are smeared out due to the longer $NO_x$ lifetime in winter.

The seasonal difference is pronounced when comparing the magnitude of the $NO_2$ concentrations in Fig. 9. In general, a smaller ratio between modelled $NO_2$ concentrations and in situ measurements is detected in winter. The ratio reaches the lowest values in January, which is consistent with the general underestimation of simulations in winter as indicated by Lin et al. (2012). For simulations with the DECSO inventory, the ratio deviating more significantly from unity in winter might be due to systematic biases in the OMI $NO_2$ observations during winter time as well. Biases in OMI $NO_2$ column densities over polluted regions are introduced by the high aerosol loading, most of which are scattering aerosols in China, as aerosols are not explicitly considered in the cloud retrieval or the AMF calculation in the operational $NO_2$ product (Castellanos et a., 2015; Chimot et al., 2016; Wang et al, 2017). The aerosols effect may be more significant in winter, due to the higher aerosol concentrations and larger solar zenith angle (Ma et al., 2013). Additionally, the DECSO algorithm is more vulnerable to biased observations as a result of the smaller number of useful observations in wintertime because of the filtering of snow-covered regions. On the contrary, the simulations with the MIX inventory show ratios ranging from 1.09 to 1.22 in the second half of the year. This may signal an overestimation of total emissions, as pointed out in Sect.3.1. In addition, the assumptions used in the MIX inventory for the distribution of monthly emissions over the year may also contributes to the bias.  For example, higher power and industrial emissions are assumed in the second half of the year due to larger industrial productions and thus power generations to meet the annual total production target (Li et al., 2017a).

### 3.3 Diurnal cycle

Figure 10 presents the diurnal variability in hourly-averaged surface $NO_2$ concentrations. The simulations with both inventories and the in situ measurements exhibit a broadly similar daily variation (r = 0.81). The distinct peak in $NO_2$

concentrations in the morning hours (around 8:00 am) and in the afternoon (around 8:00 pm) detected by the measurements has been well captured by the model, which can be attributed to increasing (traffic) emissions in the rush hours indicated by the Selected Nomenclature for sources of Air Pollution Prototype (SNAP) diurnal profiles of emissions (Menut et al., 2012) adopted in the CHIMERE model (grey line). Both simulations and measurements show a drop in $NO_2$ concentrations during

daytime with the same timing and amplitude, related to the varying chemical loss rate of $NO_2$ driven by $NO_x$ photochemistry. However, the disagreement between simulated and measured values is larger at night, which may point to problems regarding the treatment of boundary layer mixing. $NO_2$ concentrations simulated by the model cannot reproduce the observed temporal pattern at night, but present constantly high values, probably caused by unrealistically low boundary layer heights and too little vertical turbulence in the model (Bessagnet et al., 2016). This has been further confirmed by the earlier

evaluation of the diurnal cycle of trace gases as modelled by CHIMERE in Lampe et al. (2009).

We separately evaluated the model performance for the daytime period (8:00–19:00), when the pattern of diurnal variations simulated by the CHIMERE model is closer to what is observed by the in situ measurements. Not surprisingly, a larger negative slope of 0.64 is obtained for the simulation with the DECSO inventory compared to the surface observations, while the slope for the simulation with the the MIX inventory has been reduced significantly to a value of 1.1 (Table 2), due to the

tendency of overestimating $NO_2$ concentrations during night in the model. Note that the slope close to unity for the simulation with the MIX inventory during daytime does not necessarily imply a perfect emission inventory, but still indicate a potential overestimation, because we expect a slope smaller than 1 (in the range of 0.64–0.86, see section 3.1) when comparing model simulations with in situ measurements which are mainly situated in populated high-concentration areas.

## 4 Conclusions

In this work we evaluated the surface $NO_2$ concentrations from the CHIMERE CTM, driven by both satellite-derived and bottom-up emission inventories, using the measurements from the ground-based air quality monitoring network of MEP. To our knowledge, this result is the first validation of modelling $NO_2$ results with this widespread in situ network, which became recently available. Our study demonstrates the capabilities of CTMs such as CHIMERE, combined with satellite observations, to simulate $NO_2$ concentrations at the surface over China. MEP in-situ measurements can serve as a useful

dataset for evaluating model simulations, but a careful selection of measurements and scaling correction are necessary to represent the averaged $NO_2$ level over the area of a grid cell. Measurements with unexpected lower annual mean $NO_2$ concentrations in urban stations compared to those in suburban stations have been discarded from the final analysis.

The model accurately reproduces the spatial variability of annual mean $NO_2$ from in-situ measurements over China, with a spatial correlation coefficient of over 0.7. In situ measurements used in this study are expected to have a positive bias when

compared to model simulations, due to a combination of preferential placement of monitors in polluted locations and the limitation of model resolution to resolve large $NO_2$ gradients over urban areas. The estimated bias is 25% (ranging between 10% and 40%), indicated by the ratios of annual mean measurements from the background stations which is expected to

better represent the grid-cell mean to the mean of corresponding measurements from all stations for selected grid cells with over 4 stations. The bias is especially pronounced for grid cells with too few stations (less than 4 in this study) or stations located close together. Negative biases have been widely detected for mountainous and northern regions, which are most likely related to the representative issue discussed above, but model uncertainties in meteorological parameters and $NO_x$

sinks will also play a role. For other regions, a negative and positive difference has been found for the simulation with the DECSO (slope = 0.74) and the MIX (slope = 1.3) inventory respectively, suggesting an underestimation and overestimation of $NO_x$ emissions from corresponding inventories. The bias between observed and modelled concentrations was reduced significantly with the slope decreasing from 1.3 to 1.0, when the spatial distribution of $NO_x$ emissions in the DECSO inventory is applied as the spatial proxy for the MIX inventory. The reduced bias suggests an improvement of the

distribution of emissions between urban and suburban/rural areas in the DECSO inventory compared to that used in the bottom-up inventory, which shed light on addressing the spatial errors in bottom-up inventories. On the other hand, we also show that the correlation coefficient of the simulated $NO_2$ concentrations versus the in situ measurements is slightly higher in the MIX based simulations as compared to the DECSO simulations. However, this does not necessarily contradict the findings that the spatial distribution of $NO_x$ emissions is more reasonable in DECSO, considering the difference in

correlation coefficient is minor but the bias in the MIX based simulations is significant. Nevertheless, the good performance of the satellite-derived emission inventory, in particular the spatial distribution of emissions, has been confirmed by the widespread in situ measurements over China for the first time in this study. The magnitude of satellite-derived emissions show slightly negative bias by taking the negative representativity offset of in-situ measurements into account, which is attributed to biases in the OMI tropospheric $NO_2$ column densities, or representation errors introduced by the projection of

the CTM onto the measured $NO_2$ satellite footprint. In addition, satellite-derived $NO_x$ emissions succeed to detect the emission trend for the period of 2010–2015, which is consistent with that in bottom-up emissions (Liu et al., 2016a; van der A et al., 2017).

The performance of the model is comparable over seasons, with a slightly better spatial correlation in winter. This is in line with previous finding of a lower model uncertainty in winter, due to the difficulties in resolving the more active $NO_x$

photochemistry and larger concentration gradients in summer by the model. In addition, the daytime diurnal cycle has been well captured by the model. However, the disagreement between simulations and measurements is in general larger during night time, which is most likely related to the uncertainty in vertical mixing in the model. This night-time issue causes an estimated bias of about +15% in the daily mean $NO_2$ concentrations.

Note that the validation performed in this study is focused on urban areas, which may bring a systematic bias to the

conclusive statements, as discussed above. Analysis focusing on rural areas is expected in the future to give a more complete picture of the performance of CTMs with inventories. In addition, an in-depth comparison of multiple models with variable chemistry schemes (e.g., Huijnen et al., 2010) is further required to quantify the influence of chemical mechanisms on simulated $NO_2$. In order to support model validation, the introduction of additional background stations, as well as the provision of detailed information about the stations, including classification and height, would be very valuable.

## Acknowledgements

This research was funded by the MarcoPolo project of the European Union Seventh Framework Programme (FP7/2007-2013) under Grant Agreement number 606953. We acknowledge Tsinghua University for providing the MIX emission inventory. We acknowledge IPSL/LMD, INERIS and IPSL/LISA in France for providing the CHIMERE model.

## Data availability

Measurements from the ground-based air quality monitoring network of MEP were obtained from www.pm25.in. The CHIMERE model outputs are available upon request from the corresponding author.

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

**Table**

Table 1 The requirement for the minimum number of urban stations.

| Population of built-up areas/k | Area of built-up areas/km$^2$ | Minimum number of stations |
|---|---|---|
| <250 | <20 | 1 |
| 250–500 | 20–50 | 2 |
| 500–1000 | 50–100 | 4 |
| 1000–2000 | 100–200 | 6 |
| 2000–3000 | 200–400 | 8 |
| >3000 | >400 | 10 (1 per 50–60 km$^2$ ) |

Table 2 Correlation coefficient, regression slope, root mean square error (RMSE), and normalized mean error (NME) in 2015 of the simulated surface $NO_2$ concentrations driven by the DECSO 2015, the MIX Asian 2010 and the corrected DECSO emission inventory versus the corrected in situ measurements. The intercept is set to 0 when performing the regression. The unit of RMSE is μg/m$^3$.

| Category | DECSO | | | | MIX | | | | corrected DECSO | | | |
|---|---|---|---|---|---|---|---|---|---|---|---|---|
| | r | slope | RMSE | NME | r | slope | RMSE | NME | r | slope | RMSE | NME |
| Main sample | 0.73 | 0.74 | 11.6 | 0.32 | 0.85 | 1.3 | 14.8 | 0.36 | 0.72 | 1.0 | 10.5 | 0.29 |
| Main sample (daytime) | 0.76 | 0.64 | 12.1 | 0.39 | 0.84 | 1.1 | 11.3 | 0.31 | 0.76 | 0.89 | 8.8 | 0.26 |
| Unselected | 0.63 | 1.0 | 9.2 | 0.25 | 0.81 | 1.5 | 19.7 | 0.56 | 0.61 | 1.3 | 15.9 | 0.42 |
| Mountainous | 0.51 | 0.35 | 15.0 | 0.65 | 0.77 | 0.77 | 11.3 | 0.44 | 0.51 | 0.50 | 13.0 | 0.53 |
| Northern | 0.92 | 0.20 | 14.9 | 0.85 | 0.81 | 0.62 | 10.4 | 0.44 | 0.92 | 0.27 | 14.1 | 0.79 |
| < 4 Stations | 0.77 | 0.65 | 11.3 | 0.40 | 0.83 | 0.99 | 9.2 | 0.29 | 0.76 | 0.90 | 9.3 | 0.30 |
| Densely located ("L") | 0.55 | 0.74 | 13.7 | 0.53 | 0.87 | 0.78 | 7.4 | 0.25 | 0.57 | 0.99 | 15.0 | 0.52 |

**Figure**

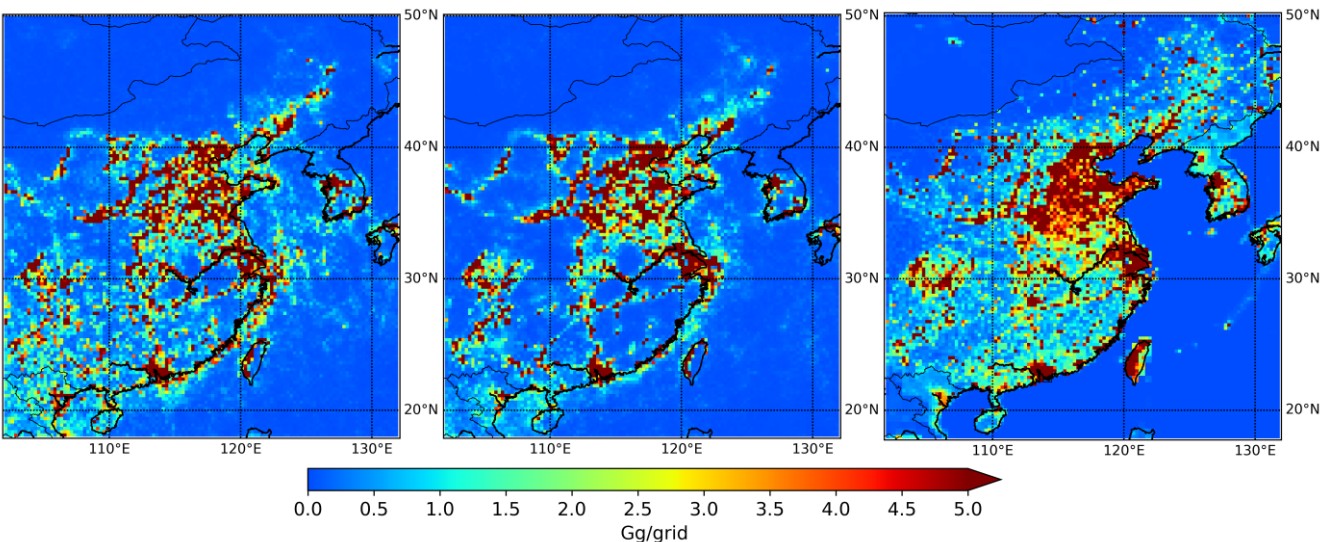

**Figure 1: Maps for NO$_x$ emissions in the DECSO inventory, 2015 (left); the DECSO inventory, 2010 (middle) and the MIX Asian inventory, 2010 (right). The unit is Gg-NO$_2$ per grid cell.**

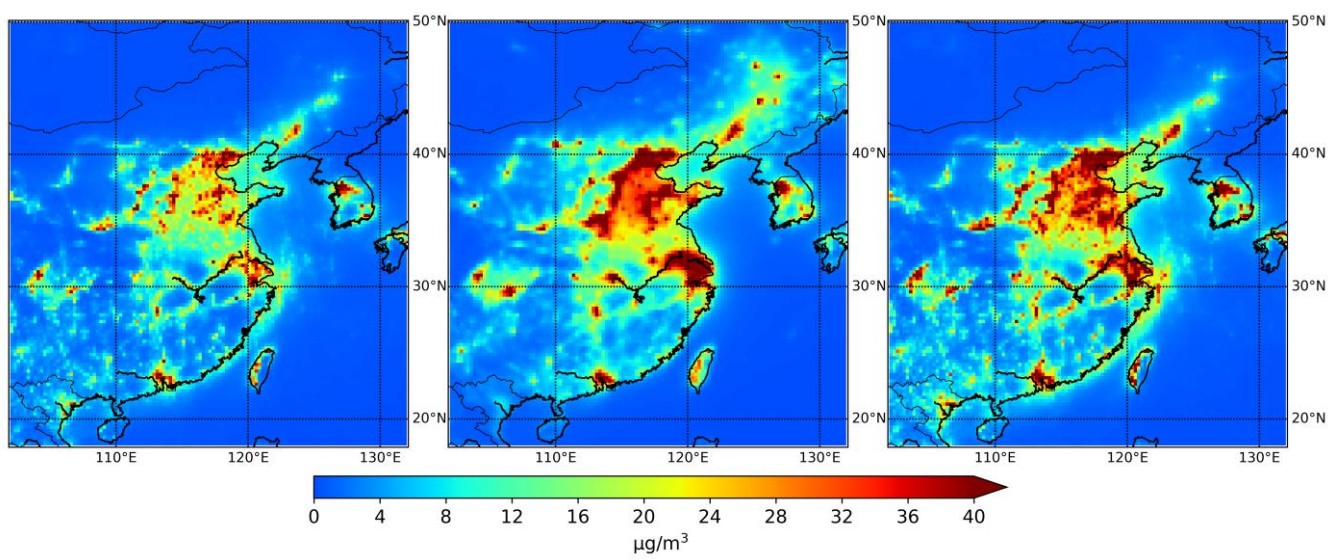

**Figure 2: Annual mean surface NO$_2$ concentration in 2015 based on the CHIMERE model driven by the DECSO inventory, 2015 (left); the MIX Asian inventory, 2010 (middle) and the corrected DECSO inventory (right).**

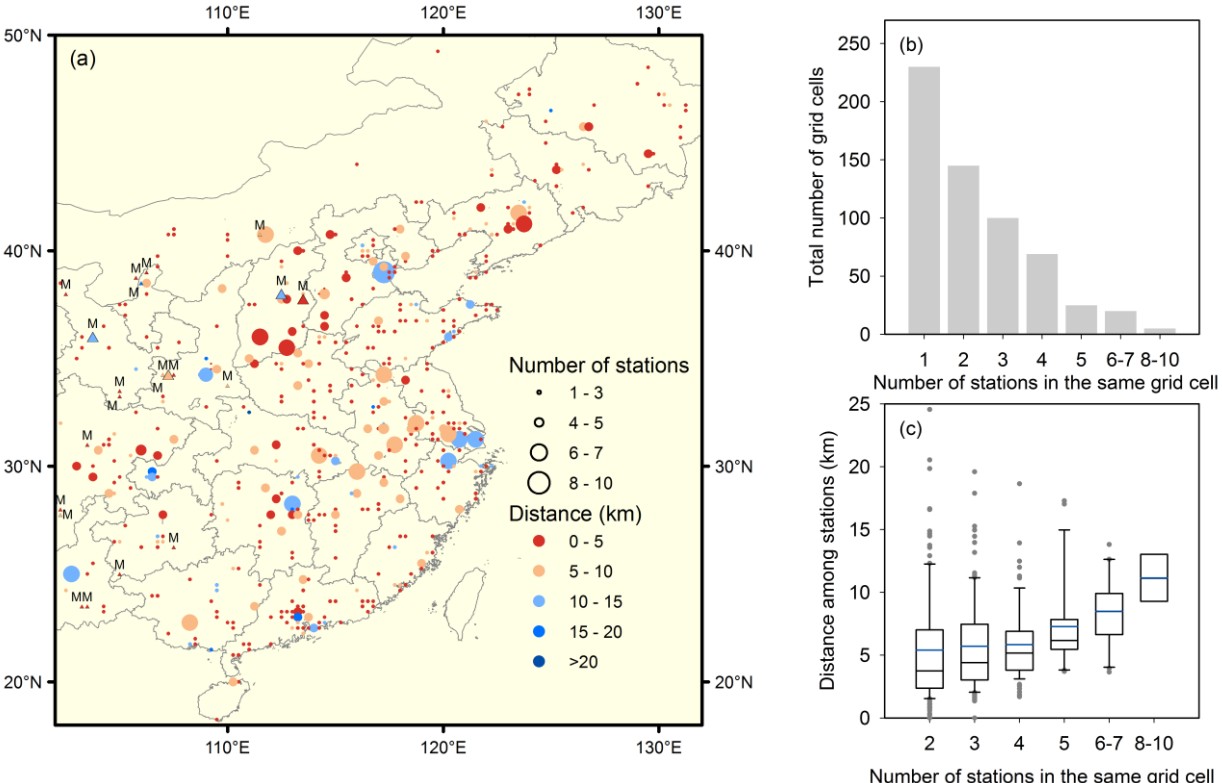

Figure 3: (a) Spatial distribution of in situ measurements. Measurements are allocated to the CHIMERE model grid cells based on their geolocations. The magnitude of the size of symbols denotes the number of stations located in the same grid cell. The colour of the symbols denotes the average distance between stations located in the same grid cell. Triangles and letters "M" denote sites located in mountainous areas. (b) Histogram of the total number of grid cells with a certain numbers of stations. (c) Statistics of the averaged distance between stations located in the same grid cell. The black and blue horizontal line is the median and mean of the averaged distance respectively; the box denotes the 25 and 75% percentiles, and the whiskers denote the 10 and 90% percentiles. The grey dots denote the outliers.

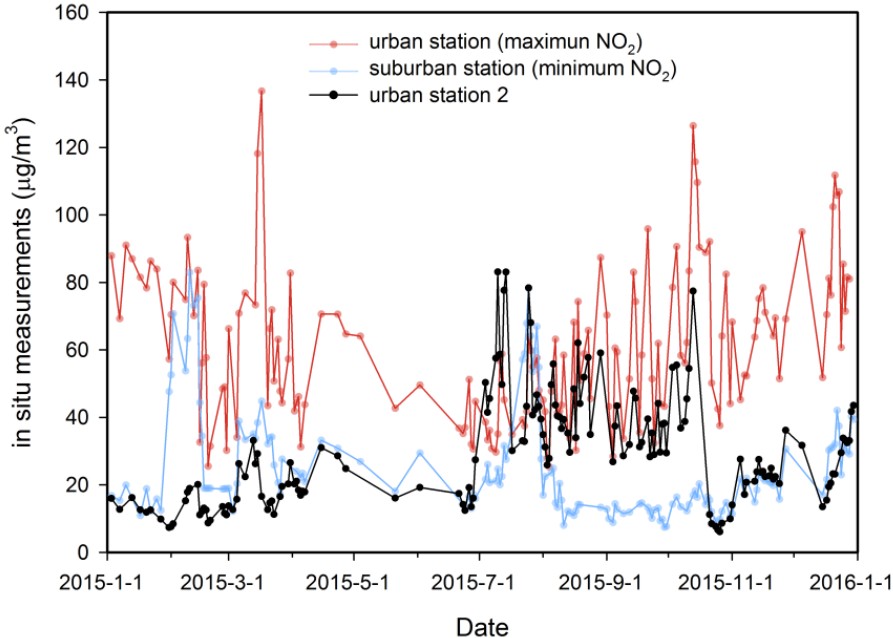

**Figure 4:** Daily mean surface $NO_2$ concentrations ($\mu g/m^3$) of stations located in the city of Xi'an for the year 2015. The data are calculated based on the measurements from the air quality monitoring network of MEP. The measurements from the stations corresponding to the maximum and minimum annual mean $NO_2$ concentrations are displayed in red and blue respectively. The measurements from the urban station with lower $NO_2$ concentrations than suburban stations are displayed in black.

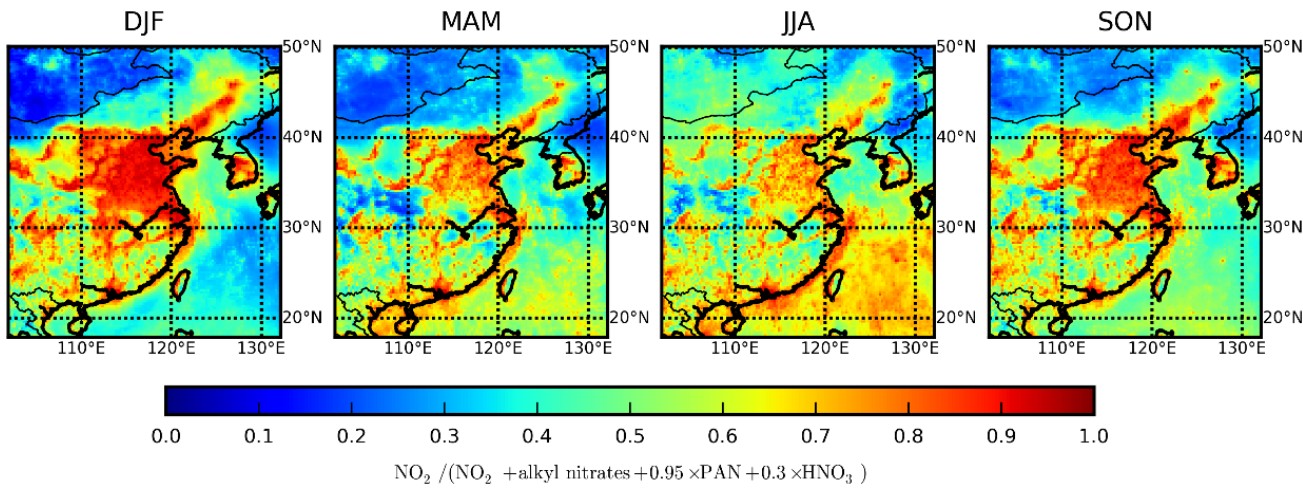

**Figure 5:** Seasonally averaged correction factors for interference in $NO_2$ measurements using chemiluminescence analyzers as estimated from a CHIMERE simulation driven by the DECSO emission inventory for the year 2015.

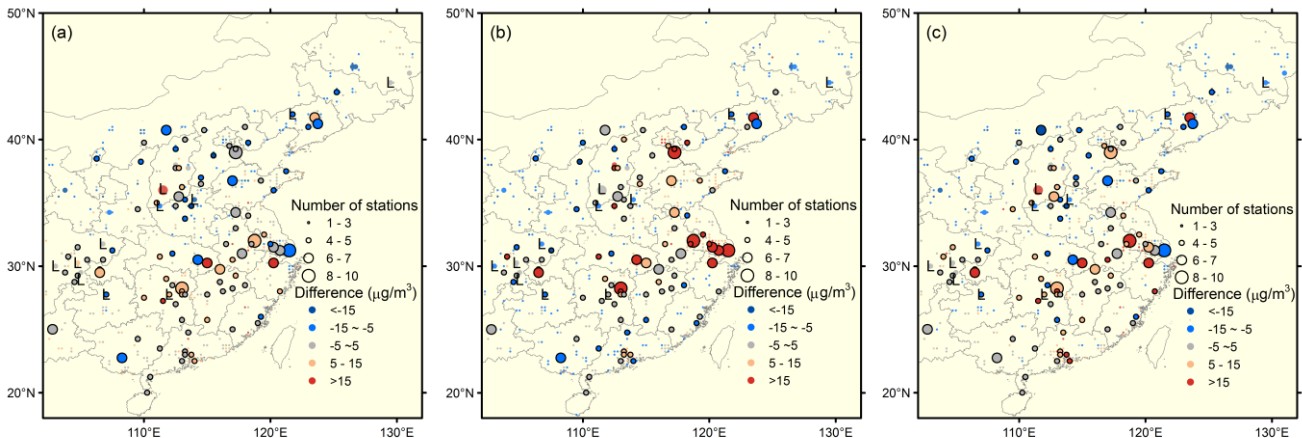

**Figure 6: The normalized difference of annual mean surface NO₂ concentrations between model simulations and the corrected in situ measurements in 2015. The simulated NO₂ concentrations driven by (a) the DECSO inventory, 2015; (b) the MIX Asian inventory, 2010 and (c) the corrected DECSO inventory are subtracted from the corrected in situ measurements to derive the differences. The mean of the differences are further subtracted from the differences to derive the normalized differences. The magnitude of the size of symbols denotes the number of stations located in the same model grid cell. The colour of the symbols denotes the difference of NO₂ concentrations. Grid cells with densely located stations are labelled with letters "L". The outline of circles corresponding to "main sample" (see Table 2) is highlighted in black.**

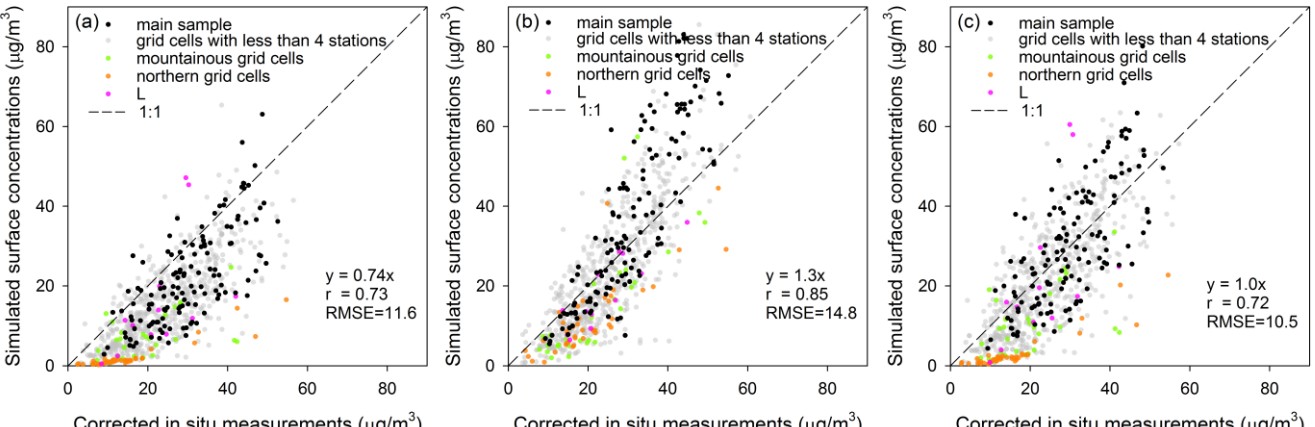

**Figure 7: Scatter plots of the simulated annual mean surface NO₂ concentrations in 2015 driven by (a) the DECSO inventory, 2015; (b) the MIX Asian inventory, 2010 and (c) the corrected DECSO inventory and the corrected in situ measurements. The orange dots correspond to the grid cells with higher latitude than 45°N. The pink dots correspond to the grid cells labelled with letters "L" in Fig. 6. The corrected DECSO inventory is derived by scaling the total amount of NOₓ emissions from the DECSO inventory to that from the MIX inventory. The intercept is set to 0 when performing the regression.**

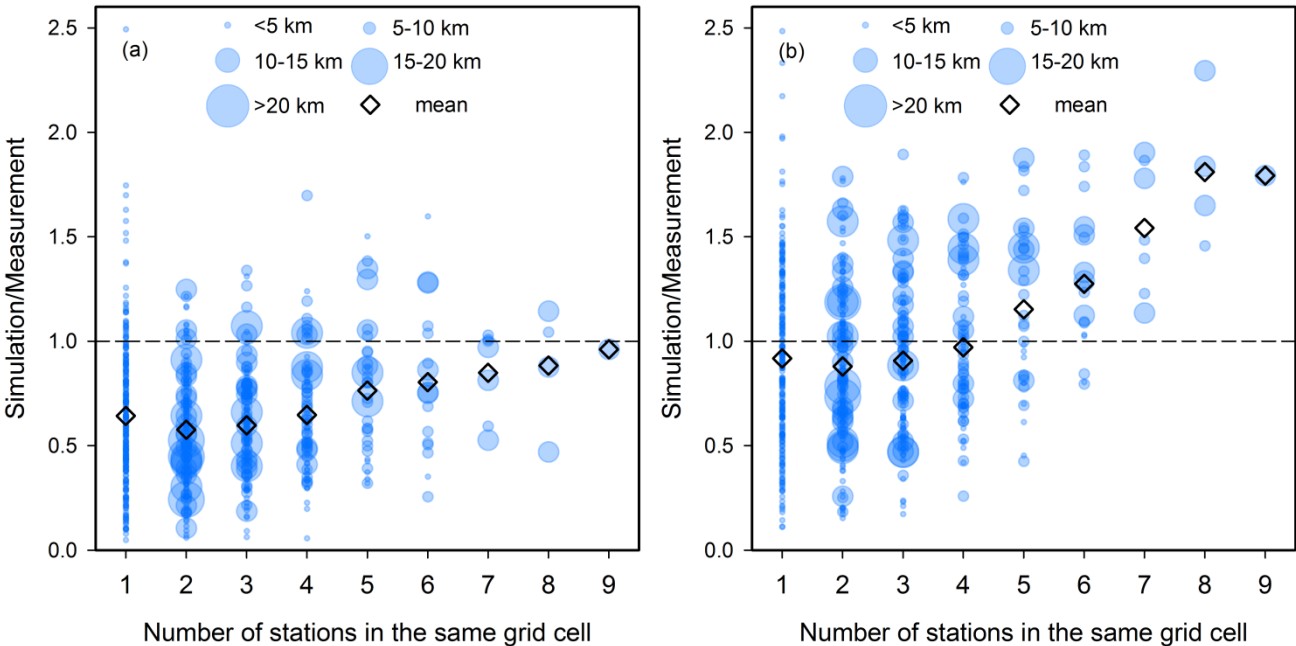

**Figure 8: Correlation among the number of stations in the same model grid cell, ratio of the simulated annual mean surface NO$_2$ concentrations in 2015 driven by (a) the DECSO inventory, 2015; (b) the MIX Asian inventory, 2010 to the corrected in situ measurements, and the average distance between stations located in the same grid cell. The magnitude of the size of the circle denotes the average distance between stations located in the same grid cell. The diamond denotes the average ratio of simulations to measurements for grid cells with different numbers of stations.**

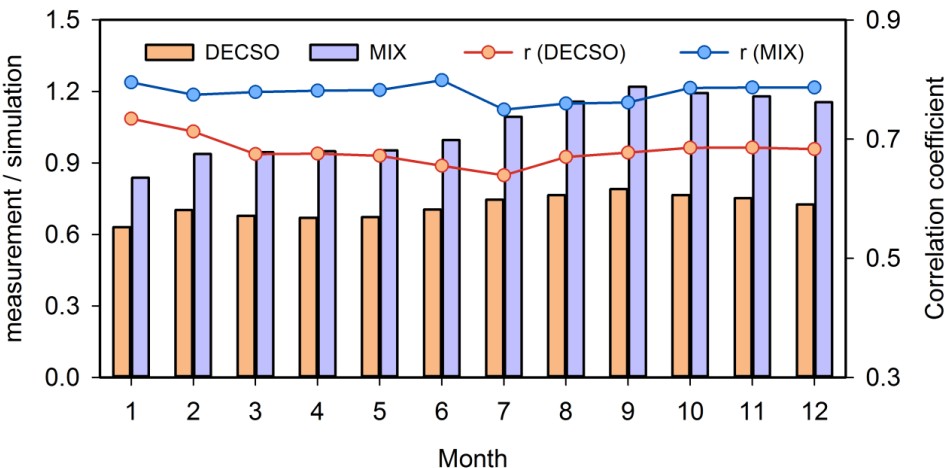

**Figure 9: Monthly mean ratio of simulated surface NO₂ concentrations driven by the DECSO inventory, 2015 (red bar) and the MIX Asian inventory, 2010 (blue bar) with the corrected in situ measurements (left axis). The correlation coefficient of the simulated NO₂ concentrations versus the corrected in situ measurements is displayed on the right axis.**

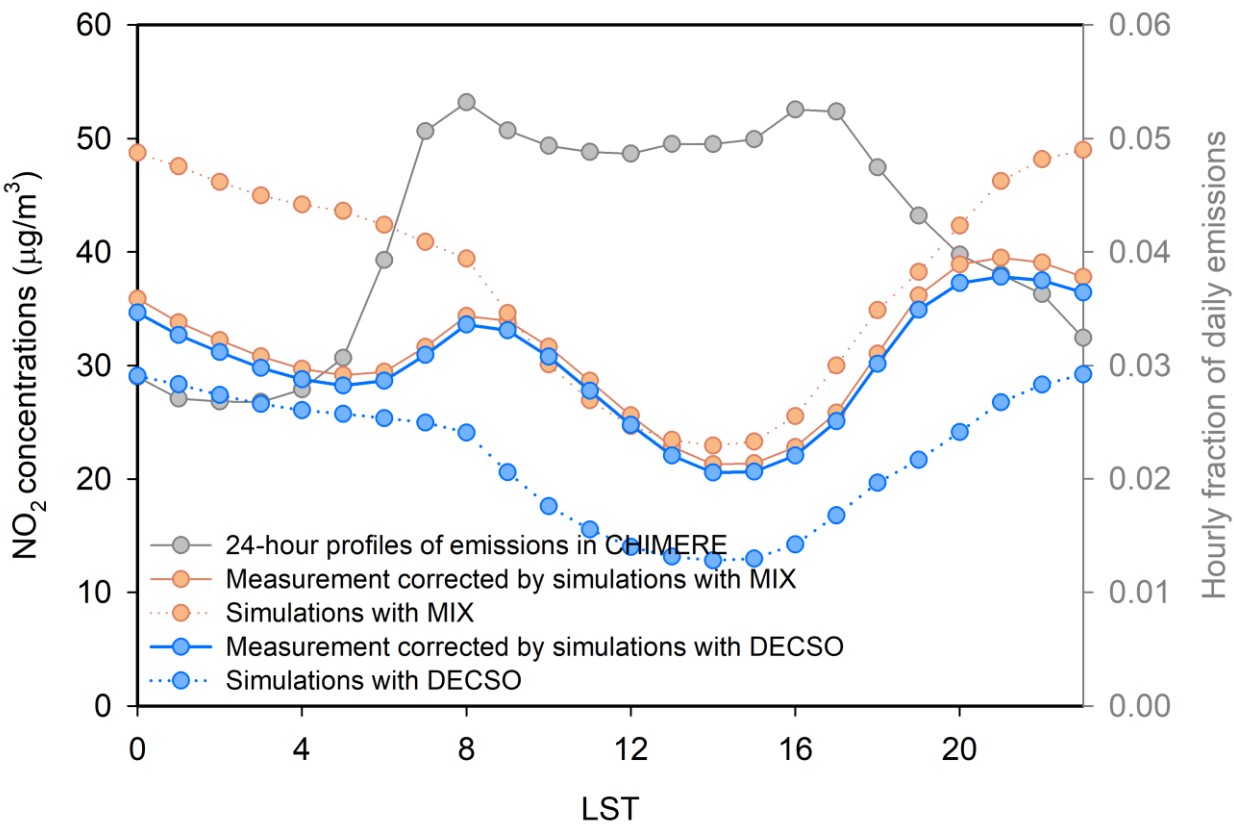

**Figure 10: Diurnal variability in hourly average NO$_2$ concentrations. Values shown are annual averages for the year 2015. Left axes: the solid line represents the corrected in situ measurement. The dotted line represents the simulation driven by the DECSO inventory, 2015 (blue) and the MIX Asian inventory, 2010 (red). Right axes: the grey line represents the 24-hour profiles applied over the daily emissions to obtain hourly data in the CHIMERE model.**