# Peer review of "Evaluation of modelling $NO_2$ concentrations driven by satellitederived and bottom-up emission inventories using in-situ measurements over China"

_Atmospheric Chemistry and Physics, 2017_

## Referee Comment (RC1) · Anonymous Referee #1 · 14 Nov 2017

This paper compares the modelling surface NO2 concentrations from a regional CTM CHIMERE driven by the satellite-derived DECSO and the bottom-up MIX emission inventories, against in-situ measurements over China. The detailed evaluation is well demonstrated, and the model performance seems sound. Nonetheless, this work lacks of impressive new findings or insights. Substantial improvements are needed before publication.

Major comments

1) Pg4, Ln26: The authors used the MIX 2010 data, and declared that it was suitable for the NOx simulations spanning over the year of 2015, as DECSO NOx emissions are

highly similar in year 2010 and 2015. This reasoning is not rigorous, if this work focuses on the comparisons of different emission inventories. Would the NOx emissions in bottom-up inventories also be highly similar in year 2010 and 2015? Furthermore, how about the changes in anthropogenic emissions of SO2, CO, VOCs and other reactive species between 2010 and 2015 both in DECSO and MIX? Both the differences in emissions for NOx (reactive species) and other species (e.g., CO, VOCs) between DECSO and MIX should impact the concentrations of NO2 through interfering with gas phase oxidation, particles through aerosol-radiation interactions and heterogeneous oxidation of NOx).

2) I am not sure whether this work focuses primarily on the comparisons of different emission inventories? If yes, the strengths and limitations of satellite-derived DECSO and the bottom-up MIX emission inventories have not been well discussed. Which emission inventory is better and recommended? Would the evaluation results change if a different model or chemical mechanism is used?

3) This paper made considerable efforts in classification of a total of 1413 air quality monitoring sites (e.g., urban sites, suburban sites, and sites share a grid cell). This information might not be that important, and should be presented briefly (or put into supporting information). A 0.25° resolution is a bit coarse in terms of mesoscale air quality simulations. If more sites appear in one cell, the errors would counteract. Would the evaluation results change if a different resolution is used?

4) Table 2 in Pg19: DECSO and MIX underestimated and overestimated the observed NOx, respectively. Then the corrected DECSO, scaled from the original DECSO using the ratio of MIX NOx to DECSO NOx over the simulation domain, simulated better results. What is the point in corrected DECSO (quite arbitrary) and its evaluation statistics? Note that in Figure 2, NOx emissions from MIX and DECSO show large discrepancies in spatial distribution especially in North China Plain.

5) Description regarding the model configuration (especially the gas phase mechanism) should be added.

Minor comments

1) Pg4, Ln3: the speciation of NOx needs further discussion. The recommended ratios are from Generation of European Emission Data for Episodes (GENEMIS), then their applicability over China? How would these ratios impact the NOx simulations?

2) Pg5, Ln16: the reviewer also uses the data from air quality monitoring sites of the MEP network. How to treat the monitoring sites with abundant and even overwhelming missing data?

3) Table 2, add the unit for RMSE. Please add NME.

4) Figure 10, please add "LST" or "UTC".

5) Pg11, Ln3: the data source for the daily profile of NOx emissions.

6) Pg11, Ln6, and also in the Abstract: discussions regarding the boundary layer mixing need more verification (figures or tables).

---

## Referee Comment (RC2) · Anonymous Referee #2 · 15 Dec 2017

This is an interesting paper focused on the evaluation of the CHIMERE predictions of NO2 using different NO2 emission estimates. It provides a test of the current MIX inventory for NO2 in terms of both magnitude and spatial distribution. The spatial distribution from a satellite based top-down inventory is also tested. The paper is able to make use of the new observations of NO2 available in China. The comparison and description of the NO2 observations is very good and this data set will be of interest to the science community. The conclusions that the MIX inventory totals for 2010 along with the updated 2015 satellite derived spatial distribution provide the most accurate prediction of 2015 surface NO2 concentrations is interesting. However the analysis is based on taking the observed NO2 concentrations and then applying, a model-based

correction to "account for" know interferences in the instrument used for the NO2 measurements. These corrections can be as large as 40% (Figure 5). The impact of this correction and uncertainty associated with using the model-based values for the correction should be discussed. What is the uncertainty in the model based values? Are there observation-based approaches that can be used to test this? There are many super sites in China now and they measure the various elements of NOy and have better direct measurements of NO2. Could such data be used to give some confidence to the scaling applied to the monitored data (at least at one or more points?

---

## Referee Comment (RC3) · Anonymous Referee #3 · 22 Dec 2017

This manuscript describes results of a regional air quality model evaluation over China using the new ground-based NO2 network that was recently installed. The paper is of scientific importance because this is the first significant publication of data from this network. The model was run with both bottom-up and top-down emission inventories.

I have two major comments concerning the paper:

1) The paper states that the NO2 monitoring stations are located at least 50 m from stationary sources and at least 10 to 100 m from roadways. I don't think these restrictions are stringent enough to use stations this close to sources to evaluate a model at quarter-degree horizontal resolution (or even a few kilometer resolution!). If information

is available concerning the distances of each station from these types of sources, the stations should be screened to use only those a more substantial distance away from the sources. Model vs. observation result may change significantly as a result of such screening.

2) The paper concludes that the satellite-based DECSO emissions are too low, and the MIX emissions are too high. But, applying the MIX total emission with the DECSO spatial distribution yielded better results. The conclusions of the paper fail to address some "bigger picture" questions. What do these results imply about emissions derived from satellite data? Are the magnitudes to be trusted? What about air quality trends from satellite data? Are they meaningful? The conclusions need to be augmented to include more general implications of the results of this study.

Minor comments:

p. 3, lines 22- 23: Annual mean simulated surface NO2 is compared....

p. 5, line 20: what concentration is the grade II air quality standard for NO2?

p. 6, lines 29-30: the correction factor is largest (ideal value of 1.0) over polluted regions, where NO2 is a larger fraction....

p. 7 line 1: Hourly correction factors derived from CHIMERE.... Are the hourly factors for each hour of each day during the year, or are they means for each hour of the day computed over a season?

p. 8, lines 32 - 33: "negative representivity" May be able to reduce this by being more restrictive on the stations used with regard to proximity of sources.

p. 9, lines 12 -13: can you give any reasons why this is the case?

p. 11, line 27: "But the modelled NO2 is generally low compared to ground measurements. This is true for DECSO, but not for MIX.

p. 12, lines 9 - 10: Did you try a simulation with the MIX spatial distribution, but scaled

to the DECSO total emission? Given the greater spatial correlation with MIX, this might be worthwhile.

---

## Author Comment (AC1) · 1 Feb 2018

*This paper compares the modelling surface NO$_2$ concentrations from a regional CTM CHIMERE driven by the satellite-derived DECSO and the bottom-up MIX emission inventories, against in-situ measurements over China. The detailed evaluation is well demonstrated, and the model performance seems sound. Nonetheless, this work lacks of impressive new findings or insights. Substantial improvements are needed before publication.*

**Response:** We thank you for the comments. To our knowledge, this study is the first validation of modelling NO$_2$ results with the widespread in situ network over China, which became recently available. The validation results reveal the bias of the spatial distribution of bottom-up NO$_x$ emissions inventories and for the first time point out that such bias can be reduced significantly by applying the spatial distribution of NO$_x$ emissions in the satellite-derived inventory as the spatial proxy for the bottom-up inventory. In addition, this study explores a reasonable scheme to classify MEP in-situ stations over China, which helps to draw more solid conclusions on the ability of the model to reproduce the measurements. The ground-based air quality monitoring network of MEP in China is a valuable database to validate model simulations, however it lacks provision of detailed information about the stations, including classification and height, and background stations which better represent regional pollution levels. Using measurements from all stations blindly without classification may bias the conclusive statements of the validation results, as the measurements may not represent the mean of a grid box of the simulations. The classification scheme developed in this study aiming to select measurements better representing grid-mean is expected to be very useful for following users of the MEP measurements.

Other reviewers have further affirmed the novelty of this study. As pointed out by other two reviewers, "The paper is of scientific importance because this is the first significant publication of data from this network." and "The comparison and description of the NO$_2$ observations is very good and this data set will be of interest to the science community. The conclusions that the MIX inventory totals for 2010 along with the updated 2015 satellite derived spatial distribution provide the most accurate prediction of 2015 surface NO$_2$ concentrations is interesting."

All the other comments have been addressed carefully below.

*Major comments*

*1) Pg4, Ln26: The authors used the MIX 2010 data, and declared that it was suitable for the NOx simulations spanning over the year of 2015, as DECSO NOx emissions are highly similar in year 2010 and 2015. This reasoning is not rigorous, if this work focuses on the comparisons of different emission inventories. Would the NOx emissions in bottom-up inventories also be highly similar in year 2010 and 2015? Furthermore, how about the changes in anthropogenic emissions of SO2, CO, VOCs and other reactive species between 2010 and 2015 both in DECSO and MIX? Both the differences in emissions for NOx (reactive species) and other species (e.g., CO, VOCs) between DECSO and MIX should impact the concentrations of $NO_2$ through interfering with gas phase oxidation, particles through aerosol-radiation interactions and heterogeneous oxidation of NOx).*

**Response:** We used the MIX inventory for 2010 instead of 2015, because the MIX inventory was developed only for the years 2008 and 2010 and this is the latest available gridded bottom-up inventories for this region. However, the usage of the 2010 inventory is not expected to bring significant bias, as the magnitude of $NO_x$ emissions for 2010 is comparable to that for 2015 in both the bottom-up inventory MEIC (Liu et al, 2016) and the satellite-derived inventory DECSO, even though there is a five-year lag. For the period of 2010–2012, the $NO_x$ emissions of China experienced a rapid growth. A sharp decline of $NO_x$ emissions was observed in the years of 2013–2015, with a peak around 2012. We have clarified this in Section 2.2 of the manuscript as follows:

"We focused on 2015 as the most recent year with available DECSO emission estimates and in situ measurements, but we used the MIX inventory for 2010, because the year 2015 is not available yet. However, the use of the 2010 MIX inventory without scaling is not expected to bring significant bias, as the similarity of $NO_x$ emissions for 2010 and 2015 has been reported by both the bottom-up inventory MEIC (Liu et al, 2016a) and the satellite-derived inventory DECSO. For the period of 2010–2012, the $NO_x$ emissions of China experienced a rapid growth. A sharp decline of $NO_x$ emissions was observed in the years of 2013–2015, with a peak around 2012 (Liu et al., 2016a). As a result, the inventory for 2010 is comparable to that for 2015, even though there is a five-year lag. Figure 1 compares DECSO $NO_x$ emissions for 2015 (left) and 2010 (middle), which are consistent in both total amount (21.5 vs 21.6 Tg) and spatial distribution (r= 0.83)."

Concerning the emissions of other species, we used the same 2010 MIX emissions in this study, as they are the most recent emissions that are public available. From 2010 to 2015, the anthropogenic emissions of $SO_2$, CO and NMVOC for China has decreased by 2%, 5% and increased by 21%, respectively (Li et al., 2017). We agree that the differences in emissions for other species between 2010 and 2015 should affect the concentrations of $NO_2$. However, it is reasonable to use the 2010 MIX inventory without scaling, because the percentages of emission changes are far less than the uncertainty ranges of emission estimates and scaling without considering changes of spatial distributions will bring larger errors and make the validation less reliable. We have added the discussion in Section 3.1 of the manuscript, as follows:

"Note that the use of the 2010 MIX emissions for other species including $SO_2$, CO and NMVOC, due to the lack of the 2015 inventory, may introduce uncertainties to the simulating $NO_2$. The anthropogenic emissions of $SO_2$, CO and NMVOC for China have been reported to decrease by 2%, 5% and increase by 21% from 2010 to 2015, respectively (Li et al., 2017b). In gas-phase chemistry, the principal sink of $NO_x$ is oxidation to $HNO_3$. The influence of the growth in NMVOC on the oxidizing power of the atmosphere is partially compensated by the reduction in CO, as CO and hydrocarbons play similar roles in depleting oxidants following the $HO_x$-$NO_x$-CO-hydrocarbon chemical mechanisms (Jacob, 2000). Additionally, $SO_2$ contributes to influence $NO_2$ concentrations by forming aerosols, concentrations of which have impact on photolysis rates and thus photochemical reaction rates associated with $NO_x$ (Mailler et al., 2017). Nevertheless, we still believe it is reasonable to use the 2010 inventory without scaling for other species, considering the emission changes are rather small compared to the uncertainties in bottom-up estimates and scaling without taking the changes of spatial distributions into account may bring larger errors to make the validation less reliable."

*2) I am not sure whether this work focuses primarily on the comparisons of different emission inventories? If yes, the strengths and limitations of satellite-derived DECSO and the bottom-up MIX emission inventories have not been well discussed. Which emission inventory is better and recommended? Would the evaluation results change if a different model or chemical mechanism is used?*

**Response:** This work aims not only to compare bottom-up and satellite-derived inventories to identify their inconsistency, but also to gain better knowledge of the reasons for the

inconsistency and then shed light on improved emission quantification approaches. Both inventories have their own strengths and limitation, which have been detailed in our previous work (Ding et al., 2017). We have summarized them in Section 2.2 of the revised manuscript, as follows:

"Both inventories show comparable spatial distributions at a national and regional scale, but distinctions between urban and rural areas (see Sect.3.1). The strength of the MIX inventory is that it includes detailed source-category information (e.g., power plant and transportation sector) for emissions, which is useful for driving atmospheric models and designing emission mitigation policies, but not included in DECSO. The advantage of the DECSO inventory is that emissions are timely updated (as soon as the satellite observations are available); while bottom-up inventories usually lag behind a few years and are outdated by the time they become available. In addition, the spatial information in DECSO is based on OMI $NO_2$ observations, while MIX relies on spatial proxies like GDP to allocate emissions due to the lack of data. An in-depth comparison between inventories has been described by Ding et al. (2017)."

It is difficult to determine which inventory is better. The model accurately reproduces the spatial variability of $NO_2$ from in-situ measurements, with a spatial correlation coefficient of over 0.7 for simulations based on both inventories, indicating the reliability of both inventories. A negative and positive bias is found for the simulation with the satellite-derived and the bottom-up inventory respectively, suggesting an underestimation and overestimation of $NO_x$ emissions from corresponding inventories. However, we do find the consistency with in-situ measurements improves when correcting the spatial distribution of $NO_x$ emissions in the bottom-up inventory by that in the satellite-derived inventory, which suggests a promising approach to improve inventories in the future. We have clarified this in the conclusion section, as follows:

"The bias between observed and modelled concentrations was reduced significantly with the slope decreasing from 1.3 to 1.0, when the spatial distribution of $NO_x$ emissions in the DECSO inventory is applied as the spatial proxy for the MIX inventory. The reduced bias suggests an improvement of the distribution of emissions between urban and suburban/rural areas in the DECSO inventory compared to that used in the bottom-up inventory, which shed light on addressing the spatial errors in bottom-up inventories."

The validation results may change, as the simulated concentrations of $NO_2$ depend on the model and chemical mechanism used. However, we don't expect fundamental changes in our

conclusions. Firstly, the CHIMERE model is considered as a state-of-the-art model and its good performance has been confirmed by many previous studies (Marécal et al., 2015). For China, the CHIMERE results are always close to the 9-model ensemble that was developed within the EU FP7 programme two collaborative research projects, MarcoPolo and Panda (available at http://www.marcopolo-panda.eu/forecast/). Second, the modelling results with the available gas phase chemical mechanisms implemented in CHIMERE (including MELCHIOR and SAPRC07) produce similar quantities of $HO_x$ radicals and ozone (Menut et al., 2013), indicating quite comparable levels of $NO_2$. Third, the positive bias of the bottom-up inventory has been confirmed by other models in certain regions of China as well, e.g., CMAQ in Jiangsu (Zhou et al., 2017) and Hebei (Zheng et al., 2017). It is a pity that we don't have the capacity to perform an in-depth comparison of multiple models with variable chemistry schemes using both inventories, but the current evidences give us confidence in the accuracy of the CHIMERE simulations.

*3) This paper made considerable efforts in classification of a total of 1413 air quality monitoring sites (e.g., urban sites, suburban sites, and sites share a grid cell). This information might not be that important, and should be presented briefly (or put into supporting information). A 0.25 resolution is a bit coarse in terms of mesoscale air quality simulations. If more sites appear in one cell, the errors would counteract. Would the evaluation results change if a different resolution is used?*

**Response:** We treat the classification scheme as a highlight of this work. As mentioned previously, the ground-based air quality monitoring network of MEP in China is a valuable database to validate model, however it lacks provision of detailed information about the stations, including classification and height, and background stations which better represent regional air pollution levels. Using measurements from all stations blindly without classification may bias the conclusive statements of the validation results, as the measurements may not represent the mean of a grid box of the simulations. The classification scheme developed in this study aiming to select measurements better representing grid-mean is expected to be very useful for following users of the MEP measurements.

Performing simulations at higher resolutions requires finer-resolution emission inventories. MIX is developed at the resolution of $0.25° \times 0.25°$ by allocating emissions using spatial proxies.

DECSO is developed at the same resolution, determined by the comparable resolution of OMI $NO_2$ observations. Finer-resolution inventories deriving from downscaling without additional information on locations of emitting facilities are expected to have significantly larger uncertainties compared to the original inventories (Zheng et al., 2017). The uncertainties will be propagated into biases of modelling results and make the validation results less reliable. Thus, we chose the resolution of $0.25° \times 0.25°$, consistent with that of MIX and DECSO, to make the validation more reliable.

*4) Table 2 in Pg19: DECSO and MIX underestimated and overestimated the observed NOx, respectively. Then the corrected DECSO, scaled from the original DECSO using the ratio of MIX NOx to DECSO NOx over the simulation domain, simulated better results. What is the point in corrected DECSO (quite arbitrary) and its evaluation statistics? Note that in Figure 2, NOx emissions from MIX and DECSO show large discrepancies in spatial distribution especially in North China Plain.*

**Response:** The aim of the scaling is to better compare the spatial distributions of the two inventories and identify the sensitivity of simulating $NO_2$ on spatial distributions of $NO_x$ emissions, considering the complicated influence of emissions on modelling $NO_2$ as pointed out by the reviewer in the previous comment. We have clarified this in Section 3.1 of the revised manuscript, as follows:

"In order to better compare the spatial distributions of the two inventories and identify the sensitivity of model performance on spatial distributions of emissions, we further evaluate the impact of the spatial distribution of emissions on simulating $NO_2$ by applying the same spatial proxy for $NO_x$ emissions in both inventories."

*5) Description regarding the model configuration (especially the gas phase mechanism) should be added.*

**Response:** We thank you for the suggestion and add the model configuration in Section 2.1, as follows:

"We use the CHIMERE model v2013b over East Asia (18°N to 50°N and 102°E to 132°E) with a resolution of 0.25° following the configuration in Ding et al. (2015). The CHIMERE simulation was driven by operational meteorological data from the European Centre for

Medium-Range Weather Forecasts (ECMWF) with a horizontal resolution of 0.25°. Atmospheric variables were simulated in 8 layers from the surface to 500 hPa. Tropospheric photochemistry is represented using the reduced MELCHIOR chemical mechanism (Derognat et al., 2003), including about 120 reactions and 44 gaseous species. Aerosol module accounting both for inorganic and organic species of primary or secondary origin is included according to Bessagnet et al. (2004). Boundary conditions for the model domain were derived from monthly mean climatology based on Model for OZone And Related chemical Tracers (MOZART) second-generation (Horowitz et al., 2003) for gases, the Laboratoire de Météorologie Dynamique Zoom – Interaction avec la Chimie et les Aérosols (LMDz-INCA; Folberth et al., 2006) for nitrate and ammonium, and the Georgia Tech/Goddard Global Ozone Chemistry Aerosol Radiation and Transport (GOCART, Ginoux et al., 2001) for other aerosols."

*Minor comments*

*1) Pg4, Ln3: the speciation of NOx needs further discussion. The recommended ratios are from Generation of European Emission Data for Episodes (GENEMIS), then their applicability over China? How would these ratios impact the NOx simulations?*

**Response:** We agree that the speciation of $NO_x$ may have impact on simulated $NO_2$. However, there is no widely accepted recommendation for the $NO_x$ speciation over China and the speciation approach is rarely mentioned by the previous air quality modelling work over China. We do find limited descriptions about the speciation, which are very close to our choice. For example, Fu et al. (2009) divides the inventory $NO_x$ into 90% NO and 10% $NO_2$. Thus we do not expect significant changes to our findings. A sensitivity study of how the speciation of $NO_x$ will influence the simulation is expected in the future, but this is out of the scope for this work.

*2) Pg5, Ln16: the reviewer also uses the data from air quality monitoring sites of the MEP network. How to treat the monitoring sites with abundant and even overwhelming missing data?*

**Response:** We are not aware of any consensus about discarding monitoring sites based on the quantity of missing data. Thus, all the measurements in the dates with 24-hour valid measurements (lager than 0) are used for the analysis in this study. We have clarified it in the Sect. 2.3 of the revised manuscript, as follows:

"Note that only the measurements for the dates with 24-hour valid measurements (larger than 0) are used for the following analysis in this study."

*3) Table 2, add the unit for RMSE. Please add NME.*

**Response:** Thanks. We have added them in the revised manuscript.

*4) Figure 10, please add "LST" or "UTC".*

**Response:** Thanks. We have added "LST" in the revised plot.

*5) Pg11, Ln3: the data source for the daily profile of NOx emissions.*

**Response:** Thanks. The data source is the Selected Nomenclature for sources of Air Pollution Prototype (SNAP) diurnal profiles (Menut et al., 2012). We have added it in the revised manuscript.

*6) Pg11, Ln6, and also in the Abstract: discussions regarding the boundary layer mixing need more verification (figures or tables).*

**Response:** We attributed the nocturnal bias to uncertainties in boundary layers based on the earlier model evaluations of CHIMERE. Supporting information including figures and tables can be derived from Lampe et al. (2009), which is out of the scope of this study. We have clarified this in Sect. 3.3 of the revised manuscript, as follows:

[revised manuscript text omitted]

---

## Author Comment (AC2) · 1 Feb 2018

*This is an interesting paper focused on the evaluation of the CHIMERE predictions of NO2 using different NO2 emission estimates. It provides a test of the current MIX inventory for NO2 in terms of both magnitude and spatial distribution. The spatial distribution from a satellite based top-down inventory is also tested. The paper is able to make use of the new observations of NO2 available in China. The comparison and description of the NO2 observations is very good and this data set will be of interest to the science community. The conclusions that the MIX inventory totals for 2010 along with the updated 2015 satellite derived spatial distribution provide the most accurate prediction of 2015 surface NO2 concentrations is interesting.*

**Response:** We thank Referee #2 for the encouraging comments. All comments and suggestions have been considered carefully and well addressed below.

*However the analysis is based on taking the observed $NO_2$ concentrations and then applying, a model-based correction to "account for" know interferences in the instrument used for the $NO_2$ measurements. These corrections can be as large as 40% (Figure 5). The impact of this correction and uncertainty associated with using the model-based values for the correction should be discussed. What is the uncertainty in the model based values? Are there observation-based approaches that can be used to test this? There are many super sites in China now and they measure the various elements of $NO_y$ and have better direct measurements of $NO_2$. Could such data be used to give some confidence to the scaling applied to the monitored data (at least at one or more points)?*

**Response:** The bias in $NO_2$ measurements from chemiluminescent analyzers due to interferences from PAN and other $NO_y$ species is significant, which can reach up to 50% (Dunlea et al., 2007). The correction approach adopted by this study has been fully validated in Lamsal et al (2008) and widely applied to correct the interferences (e.g, Bechle et al., 2013; McLinden et al., 2014). We agree that the correction has uncertainties, but not significant ones based on a literature review. It is a pity that the $NO_y$ data measured by super sites in China are not available to us. Additional validation of the uncertainty of the correction is further expected when such kind of data is available. We have added the related discussions in Sect. 2.4 of the revised manuscript, as follows:

"It is difficult to quantify the accuracy of the correction factors and errors, as the collocated measurements of other oxidized nitrogen compounds are not public available. We used the standard deviation of the daily means of correction factors within a season as a measure of its uncertainty. The average standard deviations for all sites are 10%, which are comparable to the uncertainty level pointed out by the study of McLinden et al. (2014). "

**References**

Bechle, M. J., Millet, D. B., and Marshall, J. D.: Remote sensing of exposure to $NO_2$: Satellite versus ground-based measurement in a large urban area, Atmos. Environ., 69, 345–353, 2013.

Dunlea, E. J., Herndon, S. C., Nelson, D. D., Volkamer, R. M., San Martini, F., Sheehy, P. M., Zahniser, M. S., Shorter, J. H., Wormhoudt, J. C., Lamb, B. K., Allwine, E. J., Gaffney, J. S., Marley, N. A., Grutter, M., Marquez, C., Blanco, S., Cardenas, B., Retama, A., Ramos Villegas, C. R., Kolb, C. E., Molina, L. T., and Molina, M. J.: Evaluation of nitrogen dioxide chemiluminescence monitors in a polluted urban environment, Atmos. Chem. Phys., 7, 2691–2704, doi: 10.5194/acp-7-2691-2007, 2007.

Lamsal, L. N., Martin, R. V., van Donkelaar, A., Steinbacher, M., Celarier, E. A., Bucsela, E., Dunlea, E. J., and Pinto, J. P.: Ground-level nitrogen dioxide concentrations inferred from the satellite-borne Ozone Monitoring Instrument, J. Geophys. Res., 113, D16308, doi: 10.1029/2007jd009235, 2008.

McLinden, C. A., Fioletov, V., Boersma, K. F., Kharol, S. K., Krotkov, N., Lamsal, L., Makar, P. A., Martin, R. V., Veefkind, J. P., and Yang, K.: Improved satellite retrievals of $NO_2$ and $SO_2$ over the Canadian oil sands and comparisons with surface measurements, Atmos. Chem. Phys., 14, 3637–3656, doi: 10.5194/acp-14-3637-2014, 2014.

---

## Author Comment (AC3) · 1 Feb 2018

*This manuscript describes results of a regional air quality model evaluation over China using the new ground-based NO₂ network that was recently installed. The paper is of scientific importance because this is the first significant publication of data from this network. The model was run with both bottom-up and top-down emission inventories.*

**Response:** We thank Referee #3 for the encouraging comments. All comments and suggestions have been considered carefully and well addressed below.

*I have two major comments concerning the paper:*

*1) The paper states that the NO₂ monitoring stations are located at least 50 m from stationary sources and at least 10 to 100 m from roadways. I don't think these restrictions are stringent enough to use stations this close to sources to evaluate a model at quarter-degree horizontal resolution (or even a few kilometer resolution!). If information is available concerning the distances of each station from these types of sources, the stations should be screened to use only those a more substantial distance away from the sources. Model vs. observation result may change significantly as a result of such screening.*

**Response:** We agree that stations located far away from emission sources are more suitable for model validations. However, it is pity that the information on distances of each station from different types of sources is not available to us. The general description about distance in the manuscript is derived from the placement criteria of urban assessing stations laid down in the legislation (MEP, 2013).

The ground-based air quality monitoring network of MEP in China is a valuable database to validate model simulations, however it lacks provision of detailed information about the stations, e.g., the information on distances mentioned by the review. This motivates us to explore a reasonable scheme to classify MEP in-situ stations over China, aiming to select measurements that better represent the mean of a grid box of the simulations. The category of "main sample" selected by the scheme is expected to better represent the grid-mean.

*2) The paper concludes that the satellite-based DECSO emissions are too low, and the MIX emissions are too high. But, applying the MIX total emission with the DECSO spatial*

*distribution yielded better results. The conclusions of the paper fail to address some "bigger picture" questions. What do these results imply about emissions derived from satellite data? Are the magnitudes to be trusted? What about air quality trends from satellite data? Are they meaningful? The conclusions need to be augmented to include more general implications of the results of this study.*

**Response:** We thank you for the suggestions. We have included the implications of the results of this study in the conclusion section of the revised manuscript, as follows:

"Nevertheless, the good performance of the satellite-derived emission inventory, in particular the spatial distribution of emissions, has been confirmed by the widespread in situ measurements over China for the first time in this study. The magnitude of satellite-derived emissions show slightly negative bias by taking the negative representativity offset of in-situ measurements into account, which is attributed to biases in the OMI tropospheric $NO_2$ column densities, or representation errors introduced by the projection of the CTM onto the measured $NO_2$ satellite footprint. In addition, satellite-derived $NO_x$ emissions succeed to detect the emission trend for the period of 2010–2015, which is consistent with that in bottom-up emissions (Liu et al., 2016a; van der A et al., 2017)."

Minor comments:

p. 3, lines 22- 23: Annual mean simulated surface $NO_2$ is compared....

**Response:** Thanks. We have corrected it in the revised manuscript.

p. 5, line 20: what concentration is the grade II air quality standard for $NO_2$?

**Response:** The annual mean concentration is 40 $\mu g/m^3$ in the grade II air quality standard for $NO_2$. We have added it in in the revised manuscript.

p. 6, lines 29-30: the correction factor is largest (ideal value of 1.0) over polluted regions, where $NO_2$ is a larger fraction....

**Response:** Thanks. We have corrected it in the revised manuscript.

p. 7 line 1: Hourly correction factors derived from CHIMERE.... Are the hourly factors for each hour of each day during the year, or are they means for each hour of the day computed over a season?

**Response:** They are the hourly factors for each hour of each day during the year. We have clarified this in the revised manuscript.

p. 8, lines 32 - 33: "negative representivity" may be able to reduce this by being more restrictive on the stations used with regard to proximity of sources.

**Response:** We agree that the negative bias may be reduced by using stations located more far away from emission sources. However, as mentioned previously, the information on distances of each station from different types of sources is not available to us. We have tried our best to select the measurements with better representativity of the grid-mean in this study.

p. 9, lines 12 -13: can you give any reasons why this is the case?

**Response:** The MIX inventory downscales local emissions from regional totals and distribute them to grid cells using spatial proxies (e.g., population density and GDP). However, the spatial proxies may not match the locations of the individual emitting sources, especially for industrial plants located far away from urban centres that tend to have a larger population density and GDP (Liu et al., 2017). Such a decoupling will result in an overestimation of emissions over urban areas, which has been proven by the comparison of proxy-based regional inventory with high-resolution urban inventories developed from the extensive use of information of individual emitting sources (Zheng et al., 2017). We have clarified this in the revised manuscript.

p. 11, line 27: "But the modelled $NO_2$ is generally low compared to ground measurements." This is true for DECSO, but not for MIX.

**Response:** Thanks for pointing out this. We planned to state that the mean of a grid box of the simulations is expected to be lower than in-situ measurements, due to a combination of preferential placement of monitors in polluted locations and the limitation of model resolution to resolve large $NO_2$ gradients over urban areas. However, we agree that the sentence is confusing and a repeat of the following sentence. We deleted it in the revised manuscript.

p. 12, lines 9 - 10: Did you try a simulation with the MIX spatial distribution, but scaled to the DECSO total emission? Given the greater spatial correlation with MIX, this might be worthwhile.

**Response:** The correlation coefficient is slightly higher in the MIX based simulations as compared to the DECSO simulations. However, it would be rash to conclude that the spatial distribution of MIX emissions is better based on the minor difference in the correlation coefficient (0.85 vs 0.73), ignoring the significant positive bias in the MIX based simulations. On the contrary, the spatial distribution of bottom-up emissions over China has been reported to be highly uncertain and may result in an overestimation of emissions over urban areas (e.g., Zheng et al., 2017; Liu et al., 2017). We have clarified this in the revised manuscript, as follows:

"On the other hand, we also show that the correlation coefficient of the simulated $NO_2$ concentrations versus the in situ measurements is slightly higher in the MIX based simulations as compared to the DECSO simulations. However, this does not necessarily contradict the findings that the spatial distribution of $NO_x$ emissions is more reasonable in DECSO, considering the difference in correlation coefficient is minor but the bias in the MIX based simulations is significant."

The goal of developing a scaled inventory is to better compare the spatial distributions of the two inventories and identify the sensitivity of simulated $NO_2$ on spatial distributions of $NO_x$ emissions. Comparing simulations using the inventory with the DECSO spatial distribution, but scaled to the MIX total emission, to those using MIX and DECSO has reached this goal. The comparison confirmed the reported bias in the spatial distribution of bottom-up emissions and suggested a better spatial distribution in DECSO. Meanwhile, DECSO emissions are reported to be underestimated over northern part of China (Ding et al., 2017). Thus we did not consider to construct an inventory based on MIX spatial distribution and DECSO magnitude, as we already know that the total DECSO emissions are underestimated and the spatial distribution of MIX is not very accurate.

**Reference**

Ding, J., Miyazaki, K., van der A, R. J., Mijling, B., Kurokawa, J. I., Cho, S., Janssens-Maenhout, G., Zhang, Q., Liu, F., and Levelt, P. F.: Intercomparison of $NO_x$ emission inventories over East Asia, Atmos. Chem. Phys., 17, 10125–10141, doi: 10.5194/acp-17-10125-2017, 2017.

Liu, F., Zhang, Q., Ronald, J. v. d. A., Zheng, B., Tong, D., Yan, L., Zheng, Y., and He, K.: Recent reduction in NOx emissions over China: synthesis of satellite observations and emission inventories, Environ. Res. Lett., 11, 114002, 2016.

Liu, F., Beirle, S., Zhang, Q., van der A, R. J., Zheng, B., Tong, D., and He, K.: NOx emission trends over Chinese cities estimated from OMI observations during 2005 to 2015, Atmos. Chem. Phys., 17, 9261–9275, doi: 10.5194/acp-17-9261-2017, 2017.

Ministry of Environmental Protection of the People's Republic of China (MEP): Technical regulation for selection of ambient air quality monitoring stations (on trial), available at http://kjs.mep.gov.cn/hjbhbz/bzwb/dqhjbh/jcgfffbz/201309/t20130925_260810.htm (last access: 1 May 2017), 2013 (in Chinese).

van der A, R. J., Mijling, B., Ding, J., Koukouli, M. E., Liu, F., Li, Q., Mao, H., and Theys, N.: Cleaning up the air: effectiveness of air quality policy for $SO_2$ and $NO_x$ emissions in China, Atmos. Chem. Phys., 17, 1775–1789, doi: 10.5194/acp-17-1775-2017, 2017.

Zheng, B., Zhang, Q., Tong, D., Chen, C., Hong, C., Li, M., Geng, G., Lei, Y., Huo, H., and He, K.: Resolution dependence of uncertainties in gridded emission inventories: a case study in Hebei, China, Atmos. Chem. Phys., 17, 921–933, doi: 10.5194/acp-17-921-2017, 2017.

---

## Author Response (AR2)

*Most of my questions have been well addressed except that, as the authors have tried to explain the reasons for the inconsistency in NO2 concentrations predicted between DECSO and MIX, it is still not clear if this inconsistency is mainly due to the differences in emissions for NOx or other reactive species (e.g., CO, VOCs)? This issue can be investigated preliminarily by adding extra simulations switching the photochemistry off. But this is a minor suggestion for the scope of this study.*

**Response:** We thank you for confirming that most of comments have been well addressed. Concerning the emissions of other reactive species, we used the same 2010 MIX emissions in this study, as they are the most recent emissions that are public available. For the simulation with DECSO the same 2010 MIX emissions of species other than $NO_x$ are used. Therefore, the difference between the simulations of MIX and DECSO are only a result of the different $NO_x$ emissions. We agree that CO and VOC emissions were changing between 2010 and 2015. The use of 2010 MIX inventories for CO and VOC should bring uncertainties to the simulated concentrations of $NO_2$ for 2015. However, the changes of CO and VOC emissions over China are not significant, which have been reported to decrease by 5% and increase by 21% from 2010 to 2015, respectively (Li et al., 2017). Such change rates are far less than the uncertainty ranges of CO and VOC emission estimates (~70%; Zhang et al., 2009) and even the discrepancies between estimates from different bottom-up inventories. Thus we believe the use of the 2010 MIX inventory without scaling still brings solid validation results. The related discussion is included in Section 3.1 of the manuscript, as follows:

"Note that due to the lack of the 2015 inventory, the use of the 2010 MIX emissions for other species including $SO_2$, CO and NMVOC in both the MIX and the DECSO simulations may introduce uncertainties to the simulating $NO_2$. The anthropogenic emissions of $SO_2$, CO and NMVOC for China have been reported to decrease by 2%, 5% and increase by 21% from 2010 to 2015, respectively (Li et al., 2017b). In gas-phase chemistry, the principal sink of $NO_x$ is oxidation to $HNO_3$. The influence of the growth in NMVOC on the oxidizing power of the atmosphere is partially compensated by the reduction in CO, as CO and hydrocarbons play similar roles in depleting oxidants following the $HO_x$-$NO_x$-CO-hydrocarbon chemical mechanisms (Jacob, 2000). Additionally, $SO_2$ contributes to influence $NO_2$ concentrations by

forming aerosols, concentrations of which have impact on photolysis rates and thus photochemical reaction rates associated with $NO_x$ (Mailler et al., 2017). However, the emission changes are rather small compared to the uncertainties in bottom-up estimates, which are even smaller than the discrepancies between estimates from different bottom-up inventories. Thus we believe the uncertainties arising from the use of 2010 inventory is not significant. A sensitive analysis will be further expected to quantify the influence of emissions of other species on simulated $NO_2$."

[revised manuscript text omitted]